# Splicing Machinery is Dysregulated in Pituitary Neuroendocrine Tumors and is Associated with Aggressiveness Features

**DOI:** 10.3390/cancers11101439

**Published:** 2019-09-26

**Authors:** Mari C. Vázquez-Borrego, Antonio C. Fuentes-Fayos, Eva Venegas-Moreno, Esther Rivero-Cortés, Elena Dios, Paloma Moreno-Moreno, Ainara Madrazo-Atutxa, Pablo Remón, Juan Solivera, Luiz E. Wildemberg, Leandro Kasuki, Judith M. López-Fernández, Mônica R. Gadelha, María A. Gálvez-Moreno, Alfonso Soto-Moreno, Manuel D. Gahete, Justo P. Castaño, Raúl M. Luque

**Affiliations:** 1Maimonides Institute of Biomedical Research of Cordoba (IMIBIC), 14004 Cordoba, Spain; z32vabom@uco.es (M.C.V.-B.); b22fufaa@uco.es (A.C.F.-F.); esther.rivero@imibic.org (E.R.-C.); paloma.moreno.moreno.sspa@juntadeandalucia.es (P.M.-M.); juan.solivera.sspa@juntadeandalucia.es (J.S.); mariaa.galvez.sspa@juntadeandalucia.es (M.A.G.-M.); bc2gaorm@uco.es (M.D.G.); 2Department of Cell Biology, Physiology and Immunology, University of Cordoba, 14004 Cordoba, Spain; 3Reina Sofia University Hospital (HURS), 14004 Cordoba, Spain; 4CIBER Physiopathology of Obesity and Nutrition (CIBERobn), 14004 Cordoba, Spain; 5Metabolism and Nutrition Unit, Hospital Universitario Virgen del Rocío, Instituto de Biomedicina de Sevilla (IBIS), 41013 Sevilla, Spain; evam.venegas.sspa@juntadeandalucia.es (E.V.-M.); elenadios@yahoo.es (E.D.); amadrazo-ibis@us.es (A.M.-A.); pabloj.remon.sspa@juntadeandalucia.es (P.R.); alfonsom.soto.sspa@juntadeandalucia.es (A.S.-M.); 6Service of Endocrinology and Nutrition, Reina Sofia University Hospital (HURS), 14004 Cordoba, Spain; 7Service of Neurosurgery, Reina Sofia University Hospital (HURS), 14004 Cordoba, Spain; 8Neuroendocrinology Research Center/Endocrinology Division, Medical School and Hospital Universitário Clementino Fraga Filho, Universidade Federal do Rio de Janeiro, Rio de Janeiro 21941-590, Brazil; lwildemberg@gmail.com (L.E.W.); lkasuki@yahoo.com (L.K.); mgadelha@hucff.ufrj.br (M.R.G.); 9Neuroendocrinology Division, Instituto Estadual do Cérebro Paulo Niemeyer, Rio de Janeiro 20231-092, Brazil; 10Service of Endocrinology and Nutrition, Hospital Universitario de Canarias, 38320 La Laguna, Santa Cruz de Tenerife, Spain; jlfernan@ull.es

**Keywords:** splicing, spliceosome, pituitary neuroendocrine tumors, pladienolide-B

## Abstract

Pituitary neuroendocrine tumors (PitNETs) constitute approximately 15% of all brain tumors, and most have a sporadic origin. Recent studies suggest that altered alternative splicing and, consequently, appearance of aberrant splicing variants, is a common feature of most tumor pathologies. Moreover, spliceosome is considered an attractive therapeutic target in tumor pathologies, and the inhibition of SF3B1 (e.g., using pladienolide-B) has been shown to exert antitumor effects. Therefore, we aimed to analyze the expression levels of selected splicing-machinery components in 261 PitNETs (somatotropinomas/non-functioning PitNETS/corticotropinomas/prolactinomas) and evaluated the direct effects of pladienolide-B in cell proliferation/viability/hormone secretion in human PitNETs cell cultures and pituitary cell lines (AtT-20/GH3). Results revealed a severe dysregulation of splicing-machinery components in all the PitNET subtypes compared to normal pituitaries and a unique fingerprint of splicing-machinery components that accurately discriminate between normal and tumor tissue in each PitNET subtype. Moreover, expression of specific components was associated with key clinical parameters. Interestingly, certain components were commonly dysregulated throughout all PitNET subtypes. Finally, pladienolide-B reduced cell proliferation/viability/hormone secretion in PitNET cell cultures and cell lines. Altogether, our data demonstrate a drastic dysregulation of the splicing-machinery in PitNETs that might be associated to their tumorigenesis, paving the way to explore the use of specific splicing-machinery components as novel diagnostic/prognostic and therapeutic targets in PitNETs.

## 1. Introduction

Pituitary neuroendocrine tumors (PitNETs), formerly referred to as pituitary adenomas, are more abundant than often thought, as they represent approximately 15% of all brain tumors and have an estimated prevalence that ranges from 1 in 865–2688 people [1,2]. Likewise, PitNETs have been classically considered as a benign pathology because they rarely metastasize, thus they were termed adenomas. However, the great variety of clinical behaviors accompanying these pathologies, coupled to their diverse and severe associated comorbidities and increased mortality, led the “International Pituitary Pathology Club” to propose, in a recent consensus, a reclassification of pituitary tumors and to establish the nomenclature of “pituitary neuroendocrine tumors (PitNETs)” instead of “pituitary adenomas” [3]. 

Interestingly, the vast majority of PitNETs have a sporadic origin, whereas only a small percentage (5%) is due to familial tumor syndromes [4,5]. Moreover, recent studies exploring the genomic landscape of PitNETs confirmed and extended earlier studies by reporting that all major tumor subtypes studied present a relatively low number of somatic mutations per tumor and that there are scarce recurrent mutations, none of which is commonly found across PitNET subtypes [6,7]. Overall, these genomic analyses, albeit highly informative and valuable, support the contention that mutations and purely genetic alterations alone would not fully explain PitNET tumorigenesis and, therefore, that alternative oncogenic events, including epigenetic alterations [8] or miRNAs [9,10], should be explored further to understand their actual contribution in this regard. Indeed, the primary initiating cause of PitNETs development and the possible existence of general and distinctive signatures and molecular elements in this heterogeneous pathology is still under debate [4,5,11,12,13,14]. 

In this scenario, an emerging body of evidence indicates that altered alternative splicing and its consequent outcome (i.e., the appearance of abnormal patterns of splicing and therefore, the generation of aberrant splicing variants), represents a common feature across most tumor pathologies, including PitNETs [15,16,17,18,19,20,21,22]. Alternative RNA splicing is a common post-transcriptional mechanism that provides a valuable source of biological versatility under physiological circumstances for most eukaryotic genes (>95%) [23]. The intracellular machinery that catalyzes and thereby controls the process of alternative splicing is the spliceosome, a ribonucleoproteic complex that recognizes specific sequences that determine the precise localization of the exon-intron junctions [24]. This complex machinery, organized into two systems (the major and the minor spliceosome), is comprised by structural/functional ribonucleoproteins that cooperate with splicing factors (SFs), RNA-dependent ATPase/helicases, and other regulatory proteins [25,26] in a highly dynamic fashion to finely regulate the splicing process according to positional principles demarcated by an RNA binding map to enhance or silence the exon inclusion in the mature RNA [27,28]. 

Functional alterations of this splicing-regulatory machinery can compromise the normal splicing process of an ample range of genes, thus originating the appearance of multiple, often aberrant, splicing variants, which could be directly associated with the development/progression of tumor pathologies [17,18,21,22,29]. Indeed, results from our group have demonstrated that oncogenic splicing variants from somatostatin and ghrelin systems (SST_5_TMD4/5 and In1-ghrelin) are poorly expressed in normal tissue but highly expressed in neuroendocrine tumors (NETs), including PitNETs [17,18,19,20,30,31], where they increase aggressiveness features. Based on the above, the splicing-regulatory machinery is becoming an attractive therapeutic target for tumor pathologies [32]. This is the case for pladienolide-B, a natural compound that directly targets and binds a key player in the spliceosome, SF3B1, and thereby inhibits spliceosome functions, which in turn appear to mediate the antitumor properties of this promising drug [32,33,34]. 

To date, the expression pattern and putative role of spliceosome components and SFs in the development and progression of PitNETs, as well as the potential therapeutic effects of pladienolide-B in PitNET cells, has not been reported. Accordingly, we aimed to comprehensively determine and analyze the expression levels of the spliceosome core components and a selected set of relevant SFs in the main PitNETs subtypes, i.e., non-functioning pituitary tumors (NFPTs), somatotropinomas (GHomas), corticotropinomas (ACTHomas), and prolactinomas (PRLomas), as compared to normal human pituitary gland samples (NPs). Additionally, we evaluated the potential antitumor actions of pladienolide-B in PitNET cells by evaluating key functional parameters (i.e., cell proliferation/viability and hormone secretion) in human primary PitNETs cell cultures and two models of pituitary cell lines (AtT-20 and GH3).

## 2. Results

In the present study, we analyzed simultaneously the expression levels of 42 components of the splicing machinery (12 components of major spliceosome, four components of the minor spliceosome, and 26 SFs) in different PitNETs subtypes using a dynamic quantitative real-time PCR (qPCR) microfluidic array. Specifically, we evaluated the dysregulations of these spliceosome components and SFs in an ample range of human PitNETs samples in comparison with NP-glands. Thus, we analyze 88 NFPTs, 48 GHomas, 22 ACTHomas, 7 PRLomas, and 11 NPs (cohort from Spain). Additionally, we had the opportunity to evaluate the dysregulation of the splicing machinery in a second cohort of 83 GHomas from Brazil.

### 2.1. Dysregulation of Splicing Machinery in NFPTs

Results from the dynamic qPCR microfluidic custom-made array revealed a marked dysregulation of the expression levels of several components of the splicing machinery in NFPTs compared to NPs, wherein nearly half of the elements examined were clearly downregulated (18 out of 42) or exhibited a trend to be downregulated in NFPTs (Figure 1A; Appendix A). Specifically, NFPTs showed a significant downregulation of four major spliceosome components (RNU4, RNU6, U2AF1 and U2AF2), two minor spliceosome components (RNU11 and RNU6ATAC), and 12 splicing factors (SFs) (ESRP1, PTBP1, RBM17, RBM45, SND1, SRSF1, SRSF10, SRSF3, SRSF5, SRSF9, TRA2A, and TRA2B), while only the SFs MAGOH and SRRM4 were significantly overexpressed (Figure 1A and Appendix A). Interestingly, whereas non-supervised hierarchical analysis based on the expression pattern of all spliceosome components and SFs analyzed was not able to appropriately separate NFPTs from NPs (Figure 1B), Partial Least Squares-Discriminant Analysis (PLS-DA) suggested the possibility of discriminating between NFPTs and NPs using the expression pattern of certain spliceosome components and SFs (Figure 1C). In line with this, Variable Importance in Projection (VIP) score of PLS-DA analysis indicated that SRSF9, SND1, U2AF1, SRRM4, and U2AF2 were the components with a higher capacity to discriminate between both populations (Figure 1D). 

Indeed, the expression of SRSF9, SND1, U2AF1, and U2AF2 was decreased while SRRM4 expression was increased in NFPT samples, and Receiver Operating Characteristic (ROC) curve analyses of these five components corroborated their capacity to significantly discriminate between NFPTs and NPs showing an AUC of 0.94, 0.93, 0.93, 0.80, and 0.87, respectively (Figure 1F). Although the heatmap generated with the expression of these five components did not completely segregate NFPTs and NPs, it separately grouped them by clustering together all NP samples and differentiating two subpopulations of NFPTs (P1 and P2) (Figure 1E). Of note, analysis of clinical parameters of all NFPTs included in the study revealed that the overexpression of SRRM4 was significantly associated with higher chiasmatic compression (Figure 1G). Intriguingly, when the two NFPT subpopulations (P1 and P2) were analyzed separately, the five splicing-regulatory elements mentioned above showed higher expression levels in P2 compared to P1 NFPTs (Appendix A). In addition, we compared these two subpopulations for their expression of pituitary hormones, classical receptors related with pituitary pathophysiology and clinical parameters. This revealed that mRNA levels of somatostatin receptor subtypes 1 and 5 (SST_1_ and SST_5_) were lower in NFPTs P2 compared to P1, and those of SST_3_ were higher in P2, while SST_2_ and dopamine receptor subtype 2 (D_2_: D_2_T and D_2_L) levels did not differ between P1 and P2. Finally, gene expression measurement of glycoprotein hormone alpha-subunit (CGA) and follicle-stimulating hormone (FSH) and luteinizing hormone (LH) beta subunits did not reveal any difference between P2 and P1 (Appendix A). Likewise, no differences were found regarding clinical parameters between both populations of NFPTs (Appendix A). 

On the other hand, an exhaustive analysis of the first heatmap generated (Figure 1B) showed the existence of four different subpopulations of NFPTs (P1–P4; Figure 1B). However, the analysis of relevant clinical parameters did not show in this case any significant difference between these populations (Appendix A). Variable Importance in Projection (VIP) score of PLS-DA analysis unveiled certain spliceosome components and SFs with higher capacity to discriminate between the different subpopulations of NFPTs and NPs, and the heatmaps generated were able to perfectly discriminate between the different subpopulations (P1, P2, P3, or P4) and NPs, segregating them into two perfect clusters (Appendix A). Specifically, SND1 and SRSF9 were the components with higher capacity to discriminate between NFPTs P1 and NPs; U2AF1, ESRP1, KHDRBS1, and SRRM4 discriminated between NFPTs P2 and NPs; RBM17, U2AF1, SND1, and PTBP1 discriminated between NFPTs P3 and NPs, and; SRSF1 and SND1 discriminate between NFPTs P4 y NPs (Appendix A). In addition, we compared the expression of pituitary hormones and classical receptors in these subpopulations, and the results revealed a differential mRNA expression pattern of somatostatin receptors (SSTs) and D_2_L but not of pituitary hormones or D_2_T, between the different subpopulations (Appendix A). At this point, it should be mentioned that a limitation of our study is the low number of NPs samples included in the analysis. However, it is important to take into account the difficulty to obtain this type of sample.

### 2.2. Dysregulation of Splicing Machinery in GHomas

In somatotropinomas, a clear dysregulation of splicing-regulatory components was also found compared to NPs (25 out of 42 elements altered; Figure 2A and Appendix A). Specifically, analysis of the first cohort of GHomas (C1; tumors from Spain) showed an overexpression of 6 major spliceosome components (SNRNP200, U2AF1, U2AF2, TCERG1, PRPF8, and RBM22), a downregulation of one component of the minor spliceosome (RNU11), and 18 SFs significantly overexpressed (CELF1, MAGOH, SRRM4, SPFQ, PTBP1, RAVER1, RBM17, RBM3, KHDRBS1, SRSF2, SND1, SRRM1, SRSF3, SRSF5, SRSF6, SRSF9, TIA1, and TRA2B) (Figure 2A and Appendix A). A non-supervised hierarchical analysis using the expression levels of all the splicing-machinery components was able to cluster together most NP-samples, although it did not generate a complete, perfect separation of GHomas and NPs (Figure 2B), probably due to the low number of NPs in comparison to GHomas. In contrast, PLS-DA analysis unveiled a clear segregation between GHomas and NPs based on the expression pattern of certain spliceosome components and SFs (Figure 2C). In fact, VIP score of PLS-DA analysis indicated that ribonucleoprotein PTB-binding 1 (RAVER1), RNA binding motif protein 3 (RBM3), and serine/arginine-rich splicing factor 6 (SRSF6) were the components with higher capacity to discriminate between GHomas and NPs (Figure 2D). Moreover, the heatmap generated with these three SFs was able to perfectly discriminate between GHomas and NPs, segregating them into two perfect clusters (Figure 2E).

Indeed, these three SFs (RAVER1, RBM3, and SRSF6) were significantly overexpressed in this population of GHoma samples (C1 cohort) and showed ROC curves with an AUC of 0.99, 1, and 0.98, respectively (Figure 2F). Additionally, we had the opportunity to corroborate these results in another, independent, cohort of GHomas from Brazil (cohort-2; C2). In this case, we could confirm the overexpression of RAVER1 and RBM3, but not SRSF6, in this cohort of GHomas compared to NPs (Figure 2F). Similarly, ROC curves analyses of RAVER1 and RBM3, but not of SRSF6, confirmed their capacity to discriminate between NPs with an AUC of 0.77, 0.93, and 0.59, respectively (Figure 2F). Finally, the analysis of clinically relevant parameters revealed a clear association between pre-treatment with somatostatin analogues (SSAs) and higher SRSF6 mRNA expression levels in cohort C1 (Figure 2G). Interestingly, we also found that these high levels of SRSF6 were markedly linked to a lower rate of cavernous sinus invasion (Figure 2G). In the same line, high levels of RAVER1 were also associated with less cavernous sinus invasion and also with less extrasellar growth (Figure 2G). It is also worth noting that RAVER1 expression levels were also directly correlated with the expression of the aberrant splicing variant In1-ghrelin (*r*: 0.398; *p* = 0.024) but not with native ghrelin (*r*: 0.133; *p* = 0.535).

### 2.3. Dysregulation of Splicing Machinery in ACTHomas

In corticotropinomas, qPCR array revealed a significantly dysregulation of 12 splicing machinery components, three components of major spliceosome (upregulation of RNU2 and SNRNP200, and downregulation of U2AF2), one component of the minor spliceosome (upregulation of RNU12), and eight SFs (upregulation of MAGOH, NOVA1, SPFQ, KHDRBS1, SRSF2, SNW1, TRA2B, and downregulation of ESRP1) (Figure 3A and Appendix A). Non-supervised hierarchical analysis of all splicing machinery components analyzed did not generate a clustering capable to discriminate between ACTHomas and NPs (Figure 3B), also probably due to the low number of NPs in comparison to ACTHomas. Conversely, PLS-DA analysis showed a clear separation between ACTHomas and NPs (Figure 3C), and the VIP score of PLS-DA analysis revealed that the pattern of two SFs with the highest score (MAGOH and KHDRBS1) was able to discriminate between ACTHomas and NPs (Figure 3D). In line with this, the heatmap generated with these two SFs was able to discriminate between ACTHomas and NPs, segregating them into two perfect clusters (Figure 3E). Indeed, these two SFs (MAGOH and KHDRBS1) were markedly overexpressed in ACTHomas, and ROC curves of these SFs corroborated their capacity to discriminate between ACTHomas and NPs with an AUC of 1 and 0.97, respectively (*p* < 0.0001; Figure 3F). 

Additionally, we evaluated the association of clinically relevant parameters with the expression of these two key SFs in this cohort of ACTHomas (Figure 3G). Specifically, we found that expression of MAGOH was higher in women than in men, and that MAGOH overexpression was associated with less chiasmatic compression. In fact, tumor samples showing chiasmatic compression were all men with low MAGOH expression levels (with the exception of one woman) (Figure 3F). Furthermore, ACTHomas with higher MAGOH expression levels presented higher curation rate (Figure 3G). Interestingly, MAGOH mRNA levels were directly correlated with those for the aberrant splicing variant SST_5_TMD4 (truncated somatostatin receptor type 5 with 4 transmembrane domains) (r: 0.491; *p* = 0.033) but not with the canonical receptor SST_5_ (r: 0.146; *p* = 0.552).

### 2.4. Dysregulation of Splicing Machinery in PRLomas

PRLomas also exhibited a clearly dysregulated expression pattern of spliceosomal components (12 out of 42) compared to NPs (Figure 4A and Appendix A), with a clear overexpression of three major spliceosome components (PRPF40A, PRPF8, and RBM22), a downregulation of one minor spliceosome component (RNU11), and an overexpression of eight SFs (MAGOH, SRRM4, PTBP1, RAVER1, RBM3, KHDRSB1, SRSF2, and SRSF6) (Figure 4A and Appendix A). Non-supervised hierarchical analysis did not generate a perfect clustering between PRLomas and NPs (Figure 4B). However, PLS-DA analysis neatly revealed a different expression pattern between PRLomas and NPs (Figure 4C), and VIP score analysis identified three components with high capacity to discriminate between both populations (RAVER1, MAGOH, and RNU11) (Figure 4D). Indeed, the heatmap generated with the expression of these three components produced an almost complete clustering of PRLomas and NPs (all except for one NP sample) (Figure 4E). Thus, RNU11 was found to be significantly downregulated, and RAVER1 and MAGOH were overexpressed in PRLoma samples compared to NPs, showing ROC curves with an AUC of 0.97, 1, and 0.93, respectively (Figure 4F). In this tumor type, no relevant association were found with clinical parameters or splice variants.

### 2.5. Similar Dysregulation of Specific Splicing Machinery Components in all PitNET Subtypes

A fold-change representation of the splicing machinery alterations in all PitNETs subtypes analyzed revealed a common fingerprint between all of them (Appendix A). Specifically, we found a common downregulation of two minor spliceosome components (RNU11 and RNU6ATAC) and one SF (SRSF1) and also a common overexpression of one SF (SRRM4). However, these changes did not reach statistical significance in all subtypes (Appendix A). Interestingly, SRSF1 mRNA levels were directly correlated with the aberrant splicing variant SST_5_TMD4 in NFPTs (r: 0.425; *p* = 0.049), GHomas (r: 0.532; *p* = 0.002), and ACTHomas (r: 0.509; *p* = 0.026). In contrast, this SF did not correlate with the canonical receptor SST_5_ in GHomas (r: −0.037; *p* = 0.862) or ACTHomas (r: 0.140; *p* = 0.567) but directly correlated with SST_5_ in NFPTs (r: 0.329; *p* = 0.004). Additionally, expression levels of SST_5_TMD4, but not SST_5_, were also directly associated with those of RNU11 (r: 0.392; *p* = 0.026), RNU4ATAC (r: 0.422; *p* = 0.016), and RNU6ATAC (r: 0.450; *p* = 0.010) in GHomas.

### 2.6. Effect of Pladienolide-B Treatment in PitNETs Cells

Dose-response experiments using pladienolide-B in two representative pituitary cell lines, AtT-20 corticotrophs and GH3 somatotrophs, at different times of incubation showed that lower doses of pladienolide-B (10^−9^ and 10^−11^ M) did not alter cell proliferation at any of the times tested (Figure 5A). In contrast, the 10^−7^ M dose of pladienolide-B markedly decreased cell proliferation at 24, 48 and 72 h in both cell lines (Figure 5A). We also evaluated the mRNA and protein expression levels of SF3B1 (the target of pladienolide-B) in both cell lines and the results showed that SF3B1 was highly expressed in AtT-20 than in GH3 cells (Figure 5B). Interestingly, although both cell lines showed different expression levels of SF3B1, the effect of pladienolide-B in cell proliferation was very similar in both cell lines. In the same line, treatment with pladienolide-B at 10^-9^ and 10^-11^ M did not modify cell viability in primary pituitary cell cultures of GHomas, ACTHomas, and NFPTs, but treatment with 10^−7^ M clearly reduced cell viability after 48 and 72 h of incubation in GHomas and after 72 h in ACTHomas and NFPTs (Figure 5C). Based on these results, the 10^-7^ M dose was used to evaluate the direct effect of this compound on hormone secretion. The results revealed that treatment with pladienolide-B reduced GH, but not chromogranin-A, secretion after the 24 h incubation in GHomas and NFPTs, respectively (Figure 5D).

## 3. Discussion

Evidence gathered over the last years indicates that tumor pathologies, including neuroendocrine tumors (NETs) share as a common feature the altered expression of functionally and pathologically relevant splicing variants of diverse molecules, from membrane receptors to key signaling enzymes (DLK1, GHRHR, IGF1R, EGFR, CSH2, or PTEN) [35,36,37,38]. Actually, results from our group led to the identification of previously unrecognized aberrant splicing variants from somatostatin and ghrelin systems (SST_5_TMD4/5 and In1-ghrelin) and demonstrated that these variants are overexpressed in tumors and can contribute to their oncogenesis, increasing aggressiveness and malignant features in different tumor types, including PitNETs [17,18,19,20,21,22,31,39,40]. To ascertain the potential mechanisms underlying the genesis of these tumor-related abnormal splicing events, we hypothesized that they could be linked to alterations in the machinery responsible for this process, i.e., the spliceosome and its associated SFs. In line with this notion, mutations and other functional defects in certain spliceosome components have been reported to cause diverse pathologies, including cancer [41]. Accordingly, the present study was devised to determine the pattern of expression of the splicing machinery in the main types of PitNETs and to assess the potential existence of specific alterations in spliceosome components and SFs associated to each pituitary tumor type, which may serve as future tools to guide the diagnostic/prognostic of these tumors and could provide novel actionable therapeutic targets. Indeed, results from this study demonstrate, for first time, that the splicing machinery (spliceosome and SFs) seems to be distinctly dysregulated in all PitNETs subtypes compared to NP glands, and that its modulation with a specific drug targeting SF3B1, a key player in the spliceosome function, decreases aggressiveness features in PitNET cells.

One of the main findings of this study is the discovery that the spliceosome machinery is dysregulated in a tumor subtype-dependent manner, where NFPTs, GHomas, ACTHomas, and PRLomas exhibit a differentially altered pattern of expression. Of particular interest are the results found in NFPTs, which displayed a profound downregulation of most of the components analyzed, in striking contrast with the alterations observed in functioning PitNETs (GHomas, ACTHomas, and PRLomas), wherein many components are overexpressed. In line with this, previous results have demonstrated that NFPTs typically display a distinct behavior and different expression pattern of relevant components involved in pituitary cell function, such as somatostatin receptors, in comparison with functioning PitNETs and normal tissue [42,43,44,45]. Interestingly, our results showed that the expression levels of SRSF9, SND1, U2AF1, U2AF2 and SRRM4 were able to discriminate, although not perfectly, between NFPTs and NP tissues. The absence of a perfect discrimination between both populations could be, in part, due to the well-known intrinsic heterogeneous nature of NFPTs [46], which was also evident in our bioinformatic analyses showing a clear differentiation between the four subpopulations of NFPTs (P1–P4) represented in the first heatmap generated (Figure 1B) and also between the two subpopulations of NFPTs (P1 and P2) represented in the second heatmap (Figure 1E), in terms of expression of splicing machinery components and some pathophysiologically relevant receptors (SST_1–5_ and D_2_). Nonetheless, the clear alteration of these spliceosome components found in our global cohort of NFPTs has also been observed in other tumor pathologies. Specifically, SRSF9 and SND1 have been found overexpressed in several tumor pathologies, such as breast cancer, bladder cancer, glioblastoma, melanoma, or hepatocellular carcinoma, where they have been associated with an increase in cell proliferation, invasion and poor prognosis [47,48,49,50,51]. The fact that these SFs were downregulated in NFPTs, in contrast with the overexpression observed in other pathologies, may be likely reflect the complexity, heterogeneity, and limited functional deployment of these tumors. In addition, U2AF1 is an important component of the major spliceosome that has been found frequently mutated and associated to the generation of particular splicing patterns in several pathologies, including the production of oncogenic splicing variants in cancer [52,53]. In this sense, our data unveiled a clear downregulation of U2AF1 in NFPTs compared to NPs, which might suggest that not only the mutational profile but also the expression pattern could be involved in the malignant behavior of tumor pathologies including NFPTs. Consistently, we found in NFPTs a downregulation of U2AF2, an SF that heterodimerizes with U2AF1. Although the relationship of U2AF2 with tumorigenesis is poorly studied, certain reports have demonstrated cancer-associated mutations in this SF [54], and it has also been found upregulated in lung cancer and highly metastatic hepatocellular carcinoma [55]. Remarkably, we found that SRRM4 was the only SF significantly upregulated in NFPTs, which was also associated with higher chiasmatic compression. These results compare nicely with the overexpression of SRRM4 reported in small cell lung cancers and in neuroendocrine prostate cancers, where SRRM4 was also correlated with poor patient survival [56,57]. 

On the other hand, in GHomas, a profound overexpression of three SFs—ribonucleoprotein PTB-binding 1 (RAVER1), RNA binding motif protein 3 (RBM3), and serine/arginine-rich splicing factor 6 (SRSF6)—was observed, which provided an expression pattern able to discriminate neatly between GHomas and NPs. Importantly, the altered expression pattern of RAVER1 and RBM3 was corroborated in a second, independent cohort of GHomas from Brazil. Previous results from our group have revealed that the alteration of these spliceosome components could be associated to the development of different pathological conditions. Indeed, dysregulation of RAVER1 and RBM3 has been recently related with the development of non-alcoholic fatty liver disease [58], while RAVER1 has been found to be dysregulated in patients with cardiovascular disease at higher risk of type-2 diabetes development [59]. Most importantly, additional evidence suggests that alterations in the expression level of RBM3 could be associated with advanced pathological tumor stages in lung carcinoma or with aggressive features in esophageal, colorectal or breast cancer [60,61,62,63], which reinforces the crucial role of this factor in tumor pathologies. In the case of SRSF6, the results of the first cohort analyzed demonstrated that the overexpression of this SF was associated with lower cavernous sinus invasion and with SSAs pre-treatment. Interestingly, and in line with the previous observation, the difference observed in the expression levels of SRSF6 between both cohorts of GHomas (i.e., upregulated in the Spanish cohort and no change in the Brazilian cohort) could be due to the fact that the patients from Spain, but not from Brazil, were pre-treated with somatostatin analogues before surgery, which has been shown to alter the expression pattern of key receptors in pituitary tumor samples [64]. Indeed, we also found a significant association between the pre-treatment with SSAs and higher SRSF6 mRNA expression levels in C1. Therefore, although with all required caution, these results invite us to speculate that perhaps there is a link between pre-treatment with somatostatin analogues and some beneficial effects for patient harboring GHomas in regard to some clinical symptoms such as cavernous sinus invasion, which could involve the modulation of splicing-machinery elements (i.e., upregulation of SRSF6).

In ACTHomas, our results demonstrated that the altered expression of only two SFs, MAGOH and KHDRBS1, was sufficient to fully discriminate ACTHomas from NPs. These SFs were markedly upregulated in ACTHomas, which is in accordance with the increased expression of KHDRBS1 found in gastric cancer, epithelial ovarian cancer or sacral chordomas, wherein its presence was associated with poor prognosis and aggressive characteristics [65,66]. Likewise, MAGOH has been shown to be differentially expressed in breast cancer, where it served, together with other RNA processing factors, to develop a robust stratification of breast cancer subtypes [67]. However, the presence and potential role of KHDRBS1 and MAGOH in PitNETs or NPs has not hitherto been reported. In this sense, in our cohort of ACTHomas, lower levels of MAGOH were found in men, who showed more chiasmatic compression, while higher MAGOH expression levels were associated with higher curation rate. Although these results might appear contradictory, and their potential biological significance is still far from being elucidated, our findings, together with the previous observations, prompt us to suggest that a dysbalance in some elements of the splicing machinery could be functionally related to specific pathophysiologic features of PitNETs, as it is emergently clear in other types of tumors and cancers. 

Our data in PRLomas also revealed a clear dysregulation of the splicing-machinery components. Moreover, the altered expression of several splicing-regulatory elements was able to distinguish, although not in a perfect manner, between PRLomas and NPs. The lower refinement of these models as compared to the results with other PitNET subtypes might probably be associated to the low number of PRLomas analyzed in this study, owing to the difficulty to have access to this type of samples, since dopamine agonist treatment is often highly successful in patients with PRLomas. 

Together with the identification of clearly distinct, tumor type–dependent dysregulations of the components of the splicing machinery, it is worth noting that we also pinpointed a common pattern of dysregulation of two minor spliceosome components (RNU11 and RNU6ATAC) and two SFs (SRSF1 and SRRM4) in most PitNETs, irrespective of their subtype, an observation that might be patho-physiologically relevant and thus bear future clinical potential. In particular, SRSF1 has been described to interact with many different proteins to regulate several cellular functions, including splicing, and has been found overexpressed in several types of cancer (breast and lung cancer), where it is considered a proto-oncogene [68]. Indeed, our results demonstrated that SRSF1 positively correlated with the oncogenic splicing variant SST_5_TMD4 in NFPTs, GHomas, and ACTHomas, and also that RNU11, and RNU6ATAC correlated with SST_5_TMD4 in GHomas. The fact that these spliceosome components are similarly dysregulated in all PitNETs and that these components also correlated with aberrant splicing variants, despite the high heterogeneity of these tumors, invite us to speculate about the possible existence of previously unrecognized common driver alterations in pituitary tumorigenesis, which would pave the way toward the identification of novel, common therapeutic targets based on the dysregulations of these key elements. However, further studies should be conducted to test this hypothesis.

Finally, and in line with to the previous results, our study also provides an initial, unprecedented proof-of-concept on the suitability of splicing dysregulation as a novel potential target for PitNET treatment by demonstrating that the pharmacological disruption of the splicing process with specific drugs may have antitumor effects in these neoplasms. In particular, we tested the direct in vitro effect of pladienolide-B in different PitNETs subtypes and pituitary cell lines. This compound is able to directly target a key component involved in the assembly of the spliceosome SF3B1 [69], leading to the reduction of its activity [70]. Several reports have associated pladienolide-B with antitumor properties in different cancer types [33,71,72,73], but its role in PitNETs was still unknown. For these reasons, we used this compound due to its capacity to naturally disrupt the splicing process targeting SF3B1. Thus, our results demonstrate for the first time that treatment with pladienolide-B inhibits cell viability/proliferation in all PitNETs subtypes tested and in AtT-20 and GH3 cell lines, which compares well with the reduction on cell viability and colony formation observed in HeLa cells [33] and with recent data from our group demonstrating that pladienolide-B reduced proliferation, migration and tumorspheres-formation in prostate cancer cells [33]. Interestingly, NFPTs were less sensitive to the effect of pladienolide-B compared to GHomas or ACTHomas, which is in line with previous observations in response to other pharmacological treatments in NFPTs [42,43,74]. Notably, pladienolide-B was also able to reduce GH secretion after 24 h of incubation, a relevant result since tumor hypersecretion is linked to most of the symptoms caused by GHomas.

## 4. Materials and Methods

### 4.1. Drugs and Reagents

All reagents and drugs used in this study were purchased from Sigma-Aldrich (Madrid, Spain) or Fluidigm (San Francisco, CA, USA) unless otherwise specified. Pladienolide-B was obtained from Santa Cruz Biotechnology (Heidelberg, Germany).

### 4.2. Patients, Samples, and Primary Cell Cultures

Human PitNETs samples were collected during transsphenoidal surgery from 171 patients from Spain (88 NFPTs (mean age: 58 (20–83); 40% women), 48 GHomas (mean age: 43 (21–64); 59% women; Cohort 1; C1), 22 ACTHomas (mean age: 40 (18–61); 82% women), and seven PRLomas (mean age: 43 (28–74); 29% women). Moreover, a second cohort of 83 GHomas from Brazil (Cohort 2; C2) was obtained. Additionally, 11 normal pituitary glands (NP) (mean age: 61 (44–85); 50% women) were obtained during autopsies. Each pituitary sample subtype was confirmed by expert anatomo-pathologists and by the molecular screening using qPCR, as previously described [17,42,75,76]. In all cases, samples were immediately placed in sterile cold medium (S-MEM, Gibco, Madrid, Spain; supplemented with 0.1% bovine serum albumin (BSA), 0.01% L-glutamine, 1% antibiotic-antimycotic solution, and 2.5% 4-(2-hydroxyethyl)-1-piperazineethanesulfonic acid (HEPES) after surgery and rapidly frozen and stored at −80 °C until extraction for total RNA. In a second set of experiments, PitNETs samples placed in sterile cold medium after surgery were dispersed into single cells following the methods and reagents previously described [42,76]. This study was carried out within a project approved on 27th November 2013 by our Hospital Research Ethics Committee (reference 1992), was conducted in accordance with ethical standards of the Helsinki Declaration of the World Medical Association, and written informed consent was obtained from each patient.

### 4.3. Cell Lines and Culturing

The mouse corticotrope pituitary derived cell line AtT-20/D16v-F2 (ATCC® CRL-1795™) and the rat somatotrope pituitary derived cell line GH3 (ATCC® CCL-82.1™) were used in the present study. Both cell lines were checked for mycoplasma contamination by PCR [77], cultured in Dulbecco’s Modified Eagle’s Medium (DMEM) complemented with 10% fetal bovine serum (FBS), 100 U/mL penicillin/streptomycin, 0.024 M of HEPES, and maintained at 37 °C and 5% CO2, under sterile conditions.

### 4.4. RNA Extraction, Quantification and Reverse Transcription

Total RNA from fresh tissue samples was isolated using AllPrep DNA/RNA/Protein Mini Kit followed by DNase treatment using RNase-Free DNase Set (Qiagen, Limburg, Netherlands). Total RNA concentration and purity was assessed using Nanodrop 2000 spectrophotometer (Thermo Fisher, Waltham, MA, USA), and retro-transcribed using random hexamer primers with the First Strand Synthesis Kit (Thermo Fisher).

### 4.5. Analysis of Splicing Machinery Components by a Customized qPCR Dynamic Array

As previously described [58,59], a 48.48 Dynamic Array based on microfluidic technology (Fluidigim) was used to determine the expression levels of 48 transcripts in 48 PitNETs samples, simultaneously. The specific set of primers used in this study has been previously reported by our group [58,59], and include components of the major (*n* = 12) and minor (*n* = 4) spliceosome, associated SFs (*n* = 26), and three reference genes (beta-actin (ACTB), hypoxanthine phosphoribosyltransferase 1 (HPRT1), and glyceraldehyde-3-phosphate dehydrogenase (GAPDH)), used for the normalization of gene expression levels). To control for variations in the amount of RNA used and the efficiency of the reverse-transcription reaction, the expression level of each transcript was adjusted by a normalization factor (NF) calculated with the mRNA expression levels of ACTB, HPRT1, and GAPDH using Genorm 3.3 method [78].

We performed a preamplification, exonuclease treatment, and the qPCR dynamic array following the manufacturer’s instructions. Thus, 12.5 ng of cDNA of each sample were pre-amplified using 1 μL of PreAmp Master Mix (Fluidigm) and 0.5 μL of all primers mix (500 nM) in a T100 Thermal-cycler (BioRad, Hercules, CA, USA) using the following program: 1) 2 min at 95 °C; 2) 15 s at 94 °C, and 4 min at 60 °C (14 cycles). Then, samples were treated with 2 μL of 4U/ μL Exonuclease I solution (New England BioLabs, Ipswich, MA, USA) following manufacturer’s instructions. Samples were diluted with 18 μL of TE Buffer (Thermo Scientific), and 2.7 μL were mixed with 3 μL of EvaGreen Supermix (Bio-Rad) and 0.3 μL of DNA Binding Dye Sample Loading Reagent (Fluidigm). Primers were diluted to 5 μM with 2X Assay Loading Reagent (Fluidigm). Control line fluid was charged in the chip and Prime script program was run into the IFC controller MX (Fluidigm). Finally, 5 μL of each primer and 5 μL of each sample were pipetted into their respective inlets on the chip, and the Load Mix script in the IFC controller software was run. After this program, the qPCR was run using Biomark System (Fluidigm) with the following thermal profile: 1) 1 min at 95 °C; 2) 35 cycles of denaturing (5 s at 95 °C) and annealing/extension (20 s at 60 °C); and 3) a last cycle where final products were subjected to graded temperature–dependent dissociation (60 °C to 95 °C, increasing 1 °C/3 s). Results were processed with Real-Time PCR Analysis Software 3.0 (Fluidigm).

### 4.6. RNA Isolation, Reverse Transcription, and Analysis of Gene Expression Levels by qPCR

Details of RNA extraction, quantification, reverse-transcription (RT) and qPCR using specific primers included in this study (splicing factor 3b subunit 1 (SF3B1), glycoprotein hormone alpha polypeptide (CGA), follicle stimulating hormone (FSH), luteinizing hormone (LH), somatostatin receptors (SSTs), and D_2_) have been previously reported elsewhere by our group [34,78]. It should be noted that, as previously reported and based on the stringent criteria to maximize specificity and efficiency, the qPCR technique, as applied, can be used to accurately quantify copy numbers for all human transcripts included in this study [79]. The expression level of SF3B1 in PitNETs cell lines (AtT-20 and GH3) was adjusted by a normalization factor calculated with the mRNA expression levels of ACTB, HPRT1, and GAPDH using Genorm 3.3 method [78]. However, due to the limited amount of sample available, we were only able to analyze by conventional qPCR one reference gene in the case of primary PitNETs cell cultures. In this sense, we evaluated the stability of the expression of three reference genes ACTB, HPRT1, and GAPDH in all samples using Genorm 3.3 method [80], a comprehensive tool that integrates the currently available major computational programs and found ACTB to be the most stable. Taking this into account, the expression values of CGA, FSH, LH, SSTs, and D_2_ transcripts were normalized to ACTB mRNA levels.

### 4.7. Measurement of Cell Proliferation/Viability

As previously reported [17,42,75], 10,000 cells per well (for PitNET cells) and 6000 cells per well (for cell lines) were plated in 96-well plates to measure cell proliferation/viability every 24 h until 72 h using Alamar-blue reagent (Invitrogen, Madrid, Spain). Pladienolide-B was daily refreshed after each measurement, and cell proliferation/viability was evaluated using Flex-Station III System (Molecular Devices, Sunnyvale, CA, USA).

### 4.8. Measurement of Hormone Secretion

We plated 150,000–200,000 cells per well in 24-well plates in serum-containing media. GH-secreting PitNETs cells were used to analyze the effect of pladienolide-B on GH secretion after 24 h of incubation in serum-free media. GH and chromogranin-A were measured using human commercial ELISA kit (reference numbers: EIA-3552 and EIA-4937, respectively; DRG, Mountainside, NJ), according to the manufacturer’s instructions.

### 4.9. Measurement of SF3B1 by Western Blotting

Briefly, 500,000 cells/well were cultured in 6-well plates and incubated during 48 h. After this time, proteins were extracted, separated by sodium dodecyl sulfate polyacrylamide gel electrophoresis (SDS-PAGE) and transferred to nitrocellulose membranes (Millipore, Darmstadt, Germany), as previously reported [17]. Then, blocked membranes were incubated with the primary antibody to detect SF3B1 (Abcam, Cambridge, UK; ab172634) and with appropriate secondary antibody (anti-rabbit antibody from Cell Signaling, Danvers, MA, USA), and developed using an enhanced chemiluminescence detection system (GE Healthcare, Barcelona, Spain) with dyed molecular weight markers. A densitometric analysis of the bands was carried out with ImageJ software. Proteins were normalized using total protein loading (ponceau staining).

### 4.10. Statistical Analysis

All data were evaluated for heterogeneity of variance using the Kolmogorov-Smirnov test. Statistical differences from qPCR dynamic array results were evaluated by unpaired nonparametric Mann-Whitney test and data were expressed as mean ± interquartile range. As previously reported [18,31], ROC curves were used as a tool to measure how well the expression of splicing machinery components could discriminate between different diagnostic groups. Statistical analysis of ROC curves was performed by calculating the Area Under the Curve (AUC) of each component and comparing them with the AUC of the reference line using Student’s t-test. Heatmaps and clustering analysis were performed using MetaboAnalyst 3.0 [81]. In this sense, the splicing machinery components that discriminate between PitNETs and NPs were selected following two main criteria in all cases. First, the VIP score must be higher or equal than 1.5, this value being considered as a significant value in this type of analysis. Second, and in order to perform a screening of the selected splicing machinery components by the first criteria, we chose only those that are enough to get the best hierarchical clustering in the heatmaps. Moreover, PLS-DA analysis is a statistical method close to principal components analysis (PCA) that changes the maximum variance finding by a linear regression model in a different dimension showing the best elements to discriminate between different experimental groups, in this case, normal pituitary glands and PitNETs. The splicing statistical analyses from functional assays were assessed by paired parametric t-test or one-way ANOVA test followed by Dunnett’s test for multiple comparisons, and data were expressed as mean ± SEM. Clinical correlations were assessed by unpaired nonparametric Mann-Whitney test or the Spearman test. As previously reported, to normalize values within each treatment and minimize intragroup variations in the different in vitro experiments (i.e., different age of the tissue donor or metabolic environment), the values obtained were compared with vehicle-treated controls (set at 100%). All experiments were performed in a minimum of three different primary pituitary cultures from different patients (three or four replicates per treatment per experiment), unless otherwise specified. P values ≤ 0.05 were considered statistically significant. A trend for significance was indicated when *p*-values ranged between >0.05 and <0.1. All statistical analyses were performed using GraphPad Prism 6 (GraphPad Software, La Jolla, CA, USA) or SPSS version 24.0 (SPSS, Inc., Chicago, IL, USA).

## 5. Conclusions

In summary, the present results provide novel, compelling evidence to propose that the splicing machinery is severely and distinctly dysregulated in the main subtypes of PitNETs compared to NPs and identified unique fingerprints of spliceosome components in each PitNETs subtype that can accurately discriminate between normal and tumor pituitary tissues. Furthermore, we also found several components, including SFs (SRSF1 and SRRM4) and specially two minor spliceosome components (RNU11 and RNU6ATAC), commonly dysregulated in all PitNET subtypes, which positively correlated with oncogenic splicing variants and could represent novel, more general therapeutic targets in these pathologies. These discoveries open a new window to investigate the plausible contribution of splicing dysregulation and its subsequent outcomes to pituitary tumorigenesis, and to assess the potential value of specific splicing machinery components as novel diagnostic/prognostic tools in these pathologies. Furthermore, our study unveils splicing, particularly SF3B1, as a novel actionable therapeutic point that can be targeted by Pladienolide-B to combat PitNETs.

## Figures and Tables

**Figure 1 cancers-11-01439-f001:**
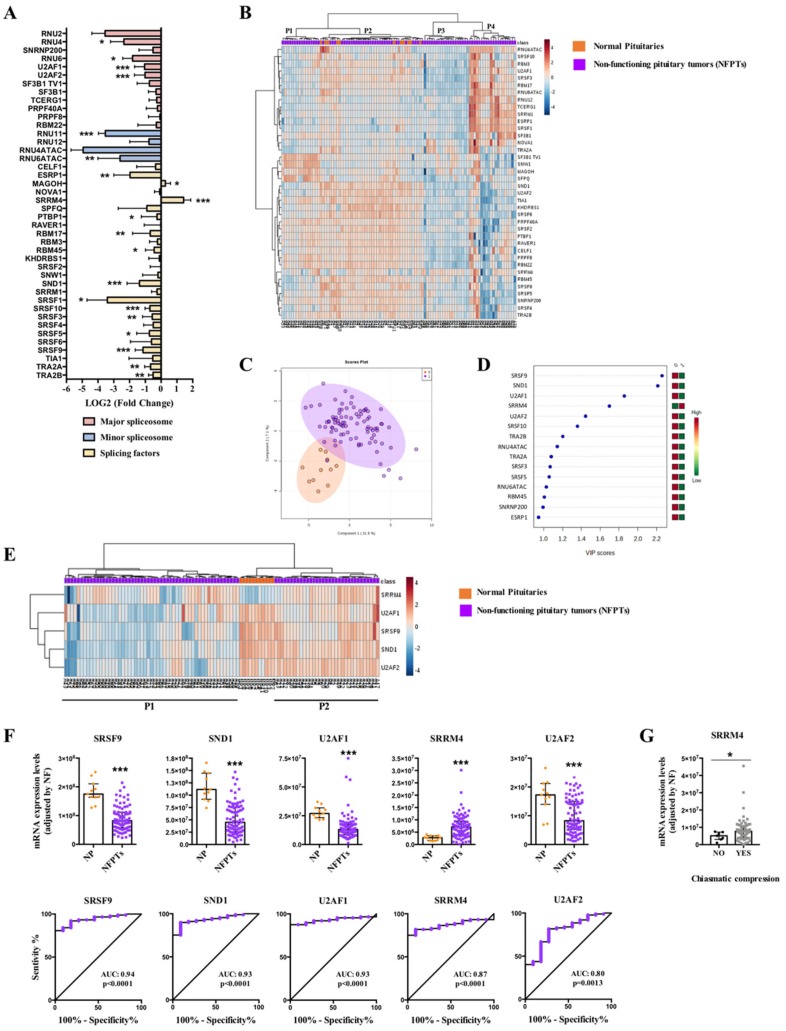
Non-functioning pituitary tumors (NFPTs). (**A**) Individual Fold-Change of each splicing-regulatory element expression levels in NFPTs compared to normal pituitary glands (NPs). (**B**) Heatmap of the mRNA expression levels of all splicing-regulatory elements measured in the qPCR array in NFPTs (*n* = 88; green color) compared to NPs (*n* = 11; red color). (**C**) Principal Components Analysis (PCA) of the mRNA expression levels of the splicing-regulatory elements analyzed in the same set of samples. (**D**) Variable Importance in Projection (VIP) Scores top-feature of Partial Least Squares Discriminant Analysis (PLS-DA). (**E**) Heatmap of the splicing-regulatory elements with higher VIP score in the same set of samples. (**F**) mRNA expression levels of splicing-regulatory elements with higher VIP score in NFPTs compared to NPs and Receiver Operating Characteristic (ROC) curves analyses showing the accuracy of the selected splicing-regulatory elements to discriminate between NFPTs and NPs. (**G**) Correlations between SRRM4 and chiasmatic compression in NFPTs. Data represent median ± interquartile range of absolute expression levels (copy number) of each transcript adjusted by a normalization factor. Asterisks (* *p* < 0.05, ** *p* < 0.01, *** *p* < 0.001) indicate statistically significant differences between groups.

**Figure 2 cancers-11-01439-f002:**
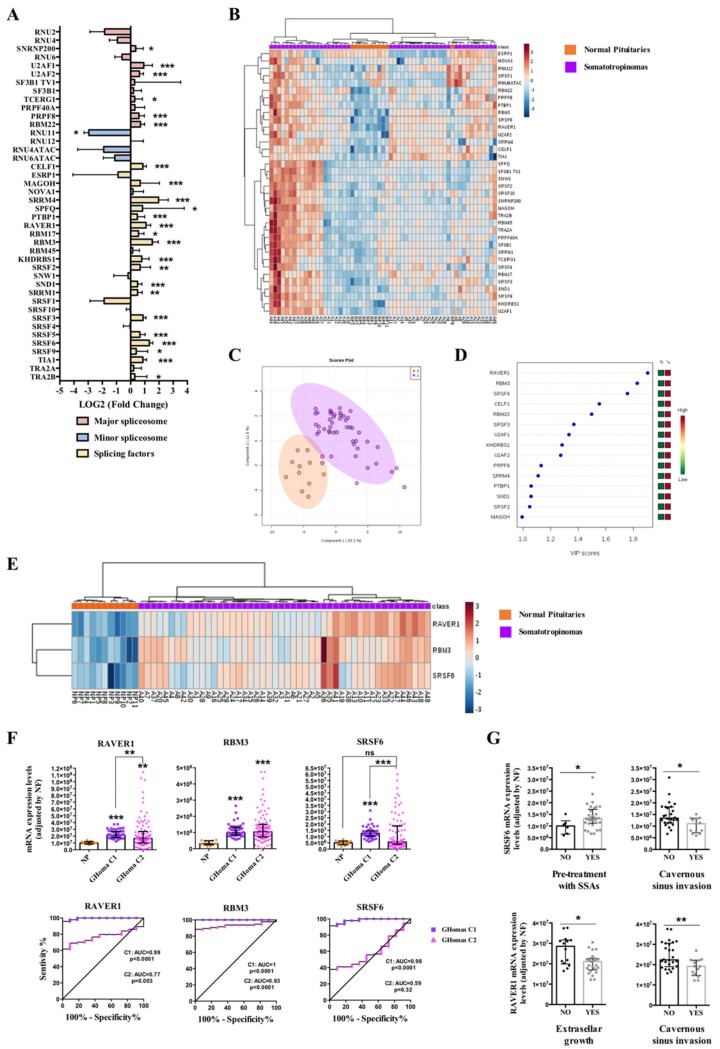
Somatotropinomas (GHomas). (**A**) Individual Fold-Change of each splicing-regulatory element expression levels in GHomas compared to normal pituitary glands (NPs). (**B**) Heatmap of the mRNA expression levels of all splicing-regulatory elements measured in the qPCR array in GHomas (n=48; green color) compared to NPs (*n* = 11; red color). (**C**) Principal Components Analysis (PCA) of the mRNA expression levels of the splicing-regulatory elements analyzed in the same set of samples. (**D**) Variable Importance in Projection (VIP) Scores top-feature of Partial Least Squares Discriminant Analysis (PLS-DA). (**E**) Heatmap of the splicing-regulatory elements with higher VIP score in the same set of samples. (**F**) mRNA expression levels of splicing-regulatory elements with higher VIP score in GHomas from cohorts 1 (C1; *n* = 48) and 2 (C2; *n* = 83) compared to NPs (*n* = 11) and Receiver Operating Characteristic (ROC) curves analyses showing the accuracy of the selected splicing-regulatory elements to discriminate between both cohorts of GHomas and NPs. (**G**) Correlations between SRSF6 and RAVER1 expression and clinical parameters. Data represent median ± interquartile range of absolute expression levels (copy number) of each transcript adjusted by a normalization factor. Asterisks (* *p* < 0.05, ** *p* < 0.01, *** *p* < 0.001) indicate statistically significant differences between groups.

**Figure 3 cancers-11-01439-f003:**
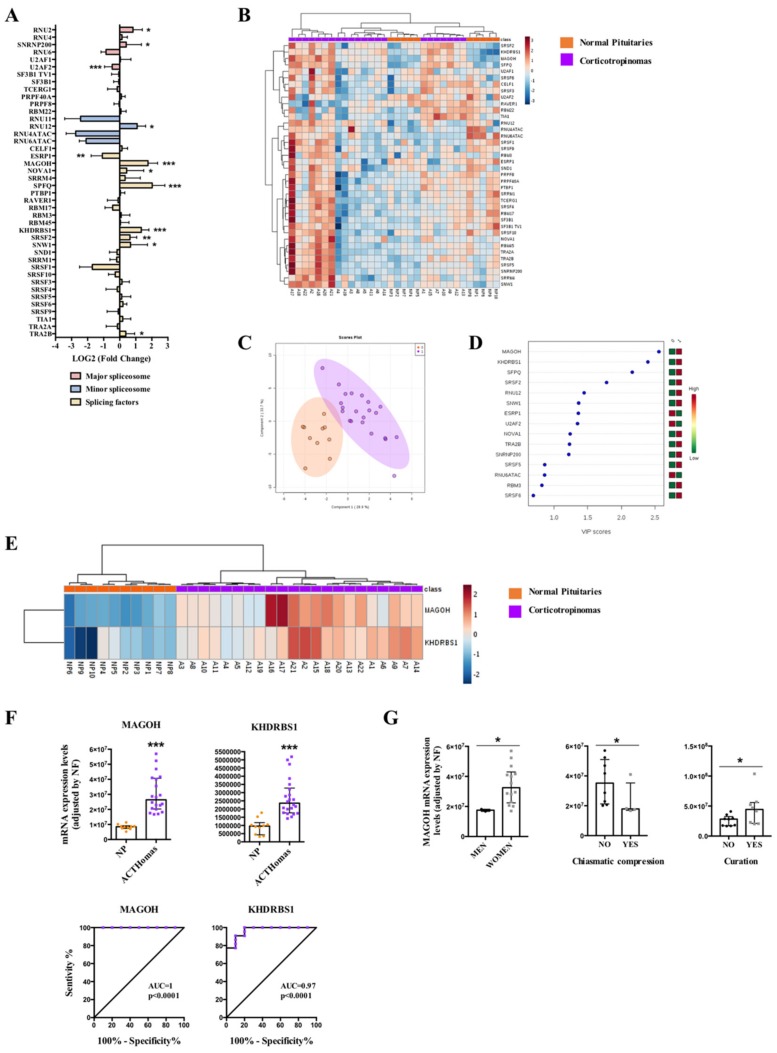
Corticotropinomas (ACTHomas). (**A**) Individual Fold-Change of each splicing-regulatory element expression levels in ACTHomas compared to normal pituitary glands (NPs). (**B**) Heatmap of the mRNA expression levels of all splicing-regulatory elements measured in the qPCR array in ACTHomas (*n* = 22; green color) compared to NPs (*n* = 10; red color). (**C**) Principal Components Analysis (PCA) of the mRNA expression levels of the splicing-regulatory elements analyzed in the same set of samples. (**D**) Variable Importance in Projection (VIP) Scores top-feature of Partial Least Squares Discriminant Analysis (PLS-DA). (**E**) Heatmap of the splicing-regulatory elements with higher VIP score in the same set of samples. (**F**) mRNA expression levels of splicing-regulatory elements with higher VIP score in ACTHomas compared to NPs and Receiver Operation Characteristic (ROC) curves analyses showing the accuracy of the selected splicing-regulatory elements to discriminate between ACTHomas and NPs. (**G**) Correlations between MAGOH and clinical parameters. Data represent median ± interquartile range of absolute expression levels (copy number) of each transcript adjusted by a normalization factor. Asterisks (* *p* < 0.05, ** *p* < 0.01, *** *p* < 0.001) indicate statistically significant differences between groups.

**Figure 4 cancers-11-01439-f004:**
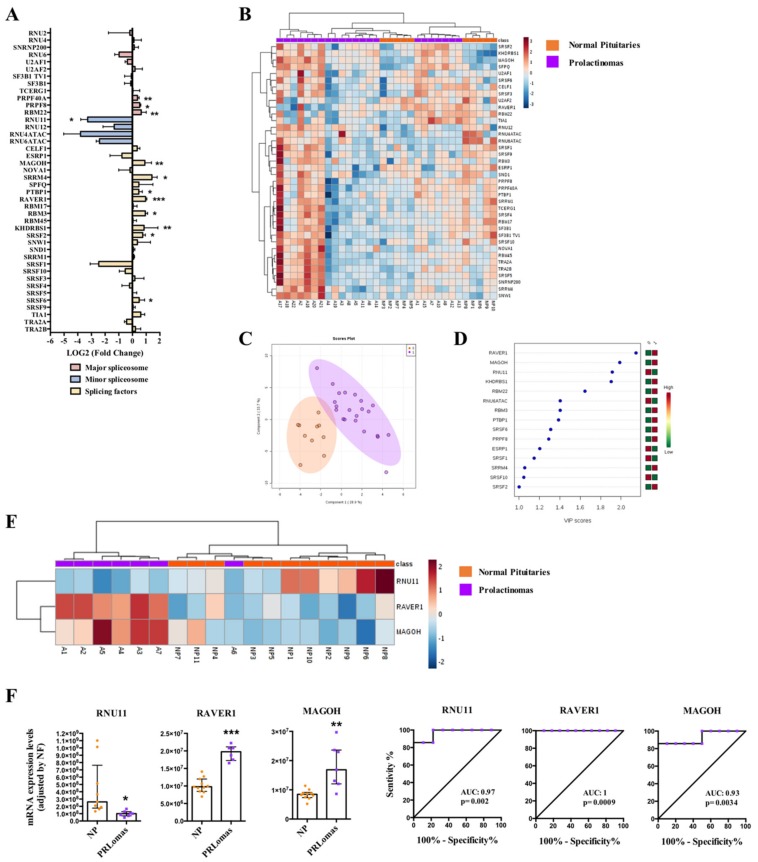
Prolactinomas (PRLomas). (**A**) Individual Fold-Change of each splicing-regulatory element expression levels in PRLomas compared to normal pituitary glands (NPs). (**B**) Heatmap of the mRNA expression levels of all splicing-regulatory elements measured in the qPCR array in PRLomas (*n* = 7; green color) compared to NPs (*n* = 11; red color). (**C**) Principal Components Analysis (PCA) of the mRNA expression levels of the splicing-regulatory elements analyzed in the same set of samples. (**D**) Variable Importance in Projection (VIP) Scores top-feature of Partial Least Squares Discriminant Analysis (PLS-DA). (**E**) Heatmap of the splicing-regulatory elements with higher VIP score in the same set of samples. (**F**) mRNA expression levels of splicing-regulatory elements with higher VIP score in PRLomas compared to NPs and Receiver Operating Characteristic (ROC) curves analyses showing the accuracy of the selected splicing-regulatory elements to discriminate between PRLomas and NPs. Data represent median ± interquartile range of absolute expression levels (copy number) of each transcript adjusted by a normalization factor. Asterisks (* *p* < 0.05; ** *p* < 0.01, *** *p* < 0.001) indicate statistically significant differences between groups.

**Figure 5 cancers-11-01439-f005:**
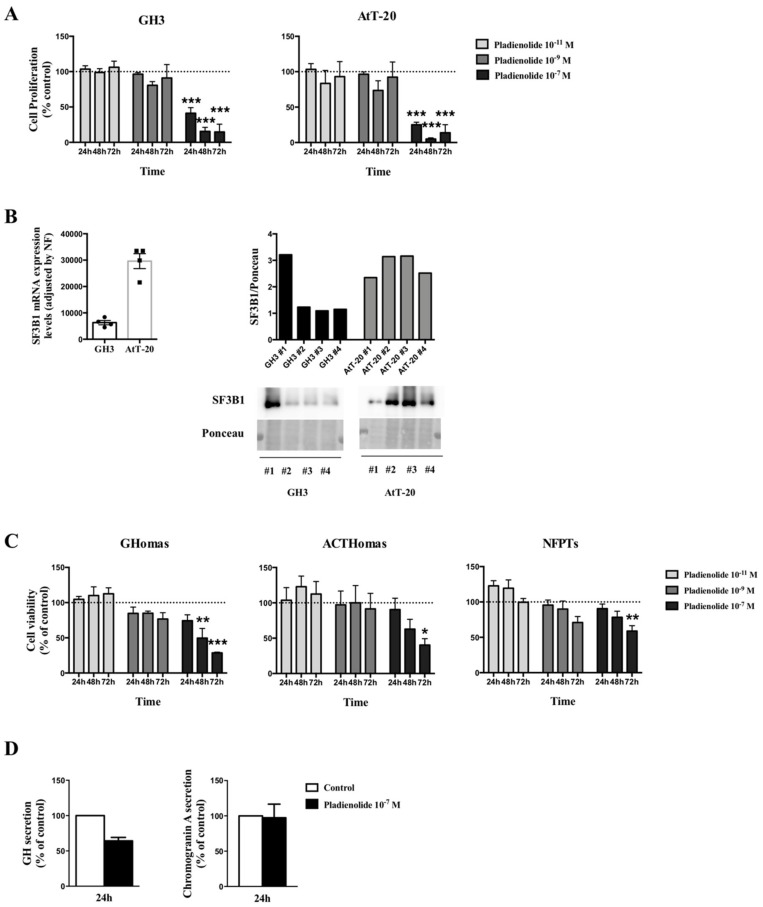
Functional assays in response to pladienolide-B in pituitary cell lines and pituitary neuroendocrine tumors (PitNETs) primary cell cultures. (**A**) Dose-response experiments of cell proliferation in response to pladienolide-B at 10^−7^, 10^−9^, and 10^−11^ M in GH3 and AtT20 cells (*n* = 4), measured by Alamar-blue reduction. (**B**) mRNA and protein levels of SF3B1 in GH3 and AtT-20 cells (*n* = 4), measured by qPCR and western blotting. (**C**) Dose-response experiments of cell viability in response to pladienolide-B in non-functioning PitNETs (NFPTs; *n* = 5), somatotropinomas (GHomas; *n* = 3), and corticotropinomas (ACTHomas; *n* = 3), measured by Alamar-blue reduction. (**D**) Effect of pladienolide-B in growth hormone (GH) and chromogranin-A secretion from GHomas and NFPTs, respectively, determined by commercial ELISA (Enzyme-Linked ImmunoSorbent Assay) kit (*n* = 2). Data are expressed as percent of vehicle-treated controls (set at 100%) within experiment. Values represent the mean ± standard error of the mean (SEM). Asterisks (* *p* < 0.05; ** *p* < 0.01; *** *p* < 0.001) indicate statistically significant differences. In cases where less than three experiments were performed, no significance tests were performed.

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
