# Peer review of "Splicing Machinery is Dysregulated in Pituitary Neuroendocrine Tumors and is Associated with Aggressiveness Features"

_cancers, 2019, doi:10.3390/cancers11101439_

Round 1

Reviewer 1 Report

The authors aimed at characterising the expression of splicing factors in pituitary neuroendocrine tumours and identifying candidates that change their expression as compared to normals. Overall, the paper is well written and easy to scientifically follow. Nevertheless, there are some aspect that need to be improved, clarified and discussed more in details to provide the reader a more comprehensive and complete view of splicing regulation in these tumors.

Here the list of my comments:

As the author mentioned (line 77), splicing is highly regulated by splicing factors, or RNA-binding proteins. The authors should also mention that the RNA-binding proteins regulate splicing according to position-dependent principles. Following those principle diverse splicing factors enhanced or silenced the inclusion of exons in the mature RNA in a cooperative o antagonist manner. The author should cite the seminal paper “Cereda et al., RNAmotifs: prediction of multivalent RNA motifs that control alternative splicing.” At line 111 and in Figure 1B, authors stated that cannot found a splicing factors that segrate NFPT and contol samples. Number of normal sample is very small as compared to NFPT. Author should mention that this is a limiting step in their analysis. Same consideration holds true for all comparisons of tumours versus normals (paragraph 2.2, 2.3). Nevertheless, it can be clearly spot from fig 1B the presence of at least 3 clusters of samples (cutting the dendrogram at second highest height). If any, what are the clinical features that distinguish these three clusters of NFPT? what are the splicing factors whose altered expression might characterised these tumours? The authors should address these points. Due to the low number of control samples, the proposed analysis might be of a more general interest. At line 114 and Figure 1C the authors selected the top 5 ranked genes for further analysis. Why the top 5? Is there a statistical motivation? How do they identified the optimal VIP cutoff? The author must explain these details and report then, even in the method section. An explanation/details on PLS-DA is also required in the Method section. At line 149 and Fig 2D, the authors again arbitrarily selected the first 3 top-ranked for this comparison. Again, what is the cutoff used for VIP analysis? These details are required At line 170 and Fig 3D, the selected splicing factors become two. Again, the authors must specify on what VIP cut-off is based their selection to allow for comparison between the analyses they performed on the different subtypes of tumours. Otherwise, these selections  sounds too arbitrary in order to justify the segregation of samples in the subsequent heatmaps (fig. 2E, 3E). Same hold true for Figure 4D/E. The authors must explain the criteria behind their choices. The authors measured cell profileferation upon the use of pladienolide-B (line 216), an inhibitor of SFSB1 (line 28). What ad the mRNA and protein expression level of SF3B1 in the two cell line models? The authors should provided these details. At lines 305 - 306  the authors states “ splicing machinery (spliceosome and SFs) is distinctly dysregulated in all PitNETs subtypes compared to NP glands “. I would suggest to lower the impact of these sentence due to the low number of NP samples used in all comparisons.

Minor:

The author must use colorblinded friendly palette. This reviewer is not colorblinded but could barely distinguish the different color (i.e. green/red) in top annotation of the heat maps. Please change them in all figures accordingly. Many acronyms are introduced but not defined (e.g. NP line 110). The authors must correct this type of issues throughout the manuscript

Author Response

RESPONSES TO REVIEWER #1:

Reviewer:As the author mentioned (line 77), splicing is highly regulated by splicing factors, or RNA-binding proteins. The authors should also mention that the RNA-binding proteins regulate splicing according to position-dependent principles. Following those principle diverse splicing factors enhanced or silenced the inclusion of exons in the mature RNA in a cooperative o antagonist manner. The author should cite the seminal paper “Cereda et al., RNAmotifs: prediction of multivalent RNA motifs that control alternative splicing”.

Authors:We thank the Reviewer for the laudatory comments and for this observation. Accordingly, we have included a sentence explaining this information and we have also included that reference in the revised manuscript. The new information can be found in page 2 (lines 74-79).“This complex machinery, organized into two systems, the major and the minor spliceosome, is comprised by structural/functional ribonucleoproteins, that cooperate with splicing factors (SFs), RNA-dependent ATPase/helicases, and other regulatory proteins in a highly dynamic fashion to finely regulate the splicing process according to positional principles demarcated by an RNA binding map to enhance or silence the exon inclusion in the mature RNA”.

Reviewer: At line 111 and in Figure 1B, authors stated that cannot found a splicing factors that segregate NFPT and control samples. Number of normal sample is very small as compared to NFPT. Author should mention that this is a limiting step in their analysis. Same consideration holds true for all comparisons of tumours versus normals (paragraph 2.2, 2.3).

Authors:Following the Reviewer’s suggestion we have mentioned in the revised manuscript that the low number of normal pituitary samples is a limitation for the analysis. However, it is important to mention the difficulty to obtain this type of samples since they are obtained during autopsies of donors. We have included this information in paragraph 2.1 (page 4, lines 161-163), 2.2 (page 4, lines 175-176) and 2.3 (page 5, lines 205-206).

Reviewer: Nevertheless, it can be clearly spot from fig 1B the presence of at least 3 clusters of samples (cutting the dendrogram at second highest height). If any, what are the clinical features that distinguish these three clusters of NFPT? what are the splicing factors whose altered expression might characterised these tumours? The authors should address these points. Due to the low number of control samples, the proposed analysis might be of a more general interest.

Authors:We thank the Reviewer for this pertinent observation. Indeed, according to the indication, we have analyzed the 4 clusters of samples (cutting the dendrogram at second highest height) taking in account clinical parameters, the expression of pituitary hormones and classical receptors associated to pituitary pathophysiology and the spliceosome components and splicing factors (SFs). The analysis of these 4 subpopulations (P1-P4) did not reveal any significant difference in the available clinical parameters that distinguish between the 4 subpopulations. Moreover, gene expression measurement of CGA, FSH and LH did not show any difference between NFPTs P1-P4. In contrast, we found a differential expression pattern of SSTs and D2 between the subpopulations, as well as a differential expression pattern of spliceosome components and SFs able to discriminate between each subpopulation of NFPTs and normal pituitaries (NPs). Interestingly, we found that at least one or more splicing machinery components from these 4 subpopulations were found altered between the splicing machinery components obtained in the global analysis of NFPTs (SRSF9, SND1, U2AF1, U2AF2 and SRRM4). Specific information of these changes have been included in pages 4 (lines 147-163), page 12 (lines 365-369) of the revised manuscript and in supplemental Table 2 and supplemental Figure 3.

Reviewer: At line 114 and Figure 1C the authors selected the top 5 ranked genes for further analysis. Why the top 5? Is there a statistical motivation? How do they identified the optimal VIP cutoff? The author must explain these details and report then, even in the method section. An explanation/details on PLS-DA is also required in the Method section. At line 149 and Fig 2D, the authors again arbitrarily selected the first 3 top-ranked for this comparison. Again, what is the cutoff used for VIP analysis? These details are required At line 170 and Fig 3D, the selected splicing factors become two. Again, the authors must specify on what VIP cut-off is based their selection to allow for comparison between the analyses they performed on the different subtypes of tumours. Otherwise, these selections sounds too arbitrary in order to justify the segregation of samples in the subsequent heatmaps (fig. 2E, 3E). Same hold true for Figure 4D/E. The authors must explain the criteria behind their choices.

Authors:We sincerely appreciate these constructive criticisms and suggestions. In this regard, the splicing machinery components that discriminate between PitNETs and NPs were selected following two main criteria in all cases. First, the VIP score must be higher or equal than 1.5, being this value considered as a significant value in this type of analysis. Second, and in order to perform a screening of the selected splicing machinery components by the first criteria, we chose only those that sufficed to obtain the best hierarchical clustering showing in Figure 1-E, 2-E, 3-E, and 4-E. The combined application of both criteria allowed to obtain a visual clustering of samples in the subsequent heatmaps. For these reasons, we have established this combination of analysis, since it provided the best option to select the splicing components that are able to discriminate between each of the different PitNETs and NPs. This information can be now found in the Statistical analysis section of the Material and Methods [page 17 (lines 580-585)].

On the other hand, PLS-DA analysis (Partial Least Squares -Discriminant Analysis; PLS-DA regression) is a statistical method close to principal components analysis (PCA) that changes the maximum variance finding by a linear regression model in a different dimension showing the best elements to discriminate between different experimental groups, in this case, normal pituitary glands and PitNETs. This information can also be found in the Statistical analysis section of the Material and Methods [page 17 (585-588)].

Reviewer: The authors measured cell proliferation upon the use of pladienolide-B (line 216), an inhibitor of SF3B1 (line 28). What ad the mRNA and protein expression level of SF3B1 in the two cell line models? The authors should provide these details.

Authors: We thank the Reviewer for this insightful observation. Following the Reviewer’s suggestion, we have measured the mRNA and protein expression levels in four different passages of GH3 and AtT-20 cells. The results showed that both cell lines express SF3B1 at mRNA and protein levels being the expression levels apparently higher in AtT-20 cells than in GH3 cells. Interestingly, although both cell lines showed different expression levels of SF3B1, the effect of pladienolide-B in cell proliferation was very similar in both cell lines. These results might suggest that the effect of pladienolide-B is not directly dependent on the expression levels of SF3B1. This new information has been included in pages 6 (lines 257-261), 16 (lines 562-570) and in figure 5.

Reviewer: At lines 305 - 306 the authors states “ splicing machinery (spliceosome and SFs) is distinctly dysregulated in all PitNETs subtypes compared to NP glands “. I would suggest to lower the impact of these sentence due to the low number of NP samples used in all comparisons.

Authors:Following the Reviewer’s suggestion we have modified the sentence from “splicing machinery (spliceosome and SFs) is distinctly dysregulated in all PitNETs subtypes compared to NP glands” to “the splicing machinery (spliceosome and SFs) seems to be distinctly dysregulated in all PitNETs subtypes compared to NP glands”, thus reducing the impact of the sentence. This correction can be found in page 12 (lines 350). Additionally, as mentioned above, we have also included the limitation of the number of NPs used in this study in several sections of the revised manuscript.

Reviewer:The author must use colorblinded friendly palette. This reviewer is not colorblinded but could barely distinguish the different color (i.e. green/red) in top annotation of the heat maps. Please change them in all figures accordingly. Many acronyms are introduced but not defined (e.g. NP line 110). The authors must correct this type of issues throughout the manuscript.

Authors: Thank you for this helpful suggestion. Following the Reviewer’s suggestion we have modified the colors of all heatmaps and graphs in the revised version of the manuscript. In the same line, we have revised and corrected the acronyms not defined.

Reviewer 2 Report

The manuscript “Splicing machinery is dysregulated in pituitary neuroendocrine tumors and is associated with aggressiveness features” by Vázquez-Borrego et al. describes aberrant expression of spliceosome components and splicing factors in different subclasses of pituitary neuroendocrine tumors (PitNETs). The widespread dysregulation of the splicing machinery has already been extensively documented in a large number of cancers. Thus, while findings on the dysregulation of the splicing machinery in cancer may translate into novel theurapeutic approaches and as such would be of interest to a broad readership, this manuscript is not sufficient to provide significant new insights. Widespread dysregulation of the splicing machinery has been amply documented in a plethora of cancer types. The authors show quantification of mRNA level changes for 42 spliceosome components. Using different clustering algorithms, they define the most significant changes, which allow separation of tumor cells from normal tissue. Further, they show correlation for some of these factors with clinical parameters. However, the authors fail to show that the identified expression changes impact on cancer cell survival. In addition, they provide no mechanistic insights, such as downstream effects on gene expression caused directly by the reduction or induction of the identified splicing factors. Therefore, I cannot recommend the manuscript for publication without further detailed mechanistic insights or showing a more far-reaching significance for this particular regulation in the progression of PitNETs.

Specific comments:

The results section should start with a more comprehensive description of the experimental setup. For example, which and how many tumor samples were compared with which and how many control samples? Of particular interest is the normalization of qPCR data. In the Material and Methods section, the authors state that three housekeeping genes were measured. Figure legends state that the data are "adjusted by a normalization factor". However, what this factor is and how normalization was performed is not described. However, this information is crucial for the evaluation and interpretation of the validity of the data. On this note, how is it possible that the same transcripts produce different results in the control compared to different PitNET subclasses (e.g. Figure S6)? The authors mention that some of the identified splicing factors are correlated with aberrant splicing variants, e.g. page 4, lines 162-164. However, the data are not shown. Figure 5: What is the difference between cell proliferation and cell viability assays? It seems to me that they are identical. Figure 5: Please add a third replicate to Figure 5C to obtain a statistically relevant result for changes in GH secretion. Discussion, page 11 line 325: SRSF9 and SND1 are overexpressed in several tumors, but reduced in NFPTs. This contradictory result should be discussed. Figure S2B: Most of the tumors do not seem to express SST1 and SST5 mRNAs at all. What percentage of tumor samples expressed these two mRNAs? Is there still a significant difference if only the samples expressing the mRNAs are considered? Why are these mRNA levels not normalized by the “normalization factor”, but by ACTB mRNA levels? Figure S6B: A significant reduction of RNU6ATAC and SRSF1 is observed only in NFPTs, but not in the other three subclasses. This can hardly be referred to as a "common fingerprint". Why, for example, was SRRM4 induction not considered? It reaches statistical significance in three out of four subclasses?

Author Response

RESPONSES TO REVIEWER #2:

Reviewer:The widespread dysregulation of the splicing machinery has already been extensively documented in a large number of cancers. Thus, while findings on the dysregulation of the splicing machinery in cancer may translate into novel therapeutic approaches and as such would be of interest to a broad readership, this manuscript is not sufficient to provide significant new insights.Widespread dysregulation of the splicing machinery has been amply documented in a plethora of cancer types.

Authors: We agree with the Reviewer in that different studies have demonstrated that certain splicing machinery components are dysregulated in several cancer types, which may help to anticipate that the general splicing machinery could be altered in a large number of cancers. However, to the best of our knowledge, the present approach, focused on the analysis of the expression of a relevant number of representative splicing machinery, and performed in parallel in a representative set of tumor samples, has not been previously reported. Moreover, as the Reviewer surely knows, tumor pathologies and cancers are very heterogeneous entities that have to be analyzed independently in order to unveil their particular features. Thus, in most studies investigating splicing in different cancer types, an analysis directed to assess the expression of the main components of the splicing machinery has not been the main focus, and thus, earlier approaches did not explore in detail the dysregulation of the splicing machinery in most cancer types. In this sense, we have implemented herein, for the first time and by the use of a pioneer, splicing machinery-focused and, custom-designed and highly sensitive approach, a comprehensive and comparative analysis of a relevant number of representative splicing machinery elements in PitNETs, demonstrating that a high proportion of these elements (including spliceosome components and splicing factors) are differentially dysregulated in different PitNETs subtypes. Therefore, we humbly consider that this study provides significant novel insights to the current state of the art and opens new conceptual pathways and methodological approaches for the study of these and other tumor pathologies.

Reviewer:The authors show quantification of mRNA level changes for 42 spliceosome components. Using different clustering algorithms, they define the most significant changes, which allow separation of tumor cells from normal tissue. Further, they show correlation for some of these factors with clinical parameters. However, the authors fail to show that the identified expression changes impact on cancer cell survival. In addition, they provide no mechanistic insights, such as downstream effects on gene expression caused directly by the reduction or induction of the identified splicing factors.

Authors: We appreciate this constructive criticism of the Reviewer. We can only concur that the impact of the splicing machinery dysregulation on cancer cell survival as well as the mechanistic insights behind this dysregulation comprise highly interesting subjects, which certainly deserve detailed investigation. Notwithstanding this, we have to point out that the main objective of our study was to implement, for the first time, a comprehensive and comparative characterization of the pattern of expression the splicing machinery components in different PitNETs subtypes compared to normal pituitaries, which unveiled a profound dysregulation of this pivotal machinery, identified novel altered targets and opened new avenues of research. In this sense, we should like to underscore that the obtaining of the high number of tumor samples from all PitNETs subtypes (being some of them very rare) analyzed in our study has been a very challenging endeavor, which required a long period of time (several years) and a tight, highly coordinated collaboration among different research centers. In addition, it should also be noted that the number of cells obtained for primary cultures after tumor dispersion is very limited, since the tumor size is usually small and the piece of tissue available for experimental studies is necessarily even smaller. For this reason, the (needless is to say) exciting and interesting study of the impact of the splicing machinery dysregulation on pituitary tumor cell survival as well as the mechanistic insights behind this dysregulation will be very challenging. Therefore, herein, as a general proof-of-concept, we had to decide to solely explore the consequences of the global inhibition of the splicing machinery by the use of pladienolide-B, which reduces SF3B1 activity and hampers the normal function of the spliceosome. Nevertheless, and in agreement with the comment by the Reviewer, we have planned to explore in the coming future the effect of the dysregulation of selected splicing machinery components on specific functional endpoints, as well as their underlying mechanistic insights. However, these planned studies will necessarily require several months and, likely, years of additional experimentation (and associated extra funding) and are meant to be the subject of independent manuscripts in the future. Accordingly, we would respectfully request to the Reviewer and Editor this additional novel and challenging data not to be considered as a requisite for the potential acceptance of our present revised manuscript.

Reviewer: The results section should start with a more comprehensive description of the experimental setup. For example, which and how many tumor samples were compared with which and how many control samples?

Authors: We thank again the Reviewer for this comment. In this regard, we have included a paragraph at the beginning of the result section explaining in detail all the samples and the splicing machinery components analyzed in our study. This information can be found in page 3 (lines 102-109). “In the present study, we analyzed simultaneously the expression levels of 42 components of the splicing machinery (12 components of major spliceosome, 4 components of the minor spliceosome and 26 SFs) in different PitNETs subtypes using a dynamic quantitative real-time PCR (qPCR) microfluidic array. Specifically, we evaluated the dysregulations of these spliceosome components and SFs in an ample range of human PitNETs samples in comparison with NP-glands. Thus, we analyze 88 NFPTs, 48 GHomas, 22 ACTHomas, 7 PRLomas and 11 NPs (Cohort from Spain). Additionally, we had the opportunity to evaluate the dysregulation of the splicing machinery in a second cohort of 83 GHomas from Brazil.”.

Reviewer:Of particular interest is the normalization of qPCR data. In the Material and Methods section, the authors state that three housekeeping genes were measured. Figure legends state that the data are "adjusted by a normalization factor". However, what this factor is and how normalization was performed is not described. However, this information is crucial for the evaluation and interpretation of the validity of the data. On this note, how is it possible that the same transcripts produce different results in the control compared to different PitNET subclasses (e.g. Figure S6)?

Authors: We sincerely appreciate this pertinent observation and apologize for not being clearer in this aspect. Specifically,the normalization of the expression of all the transcripts determined in the qPCR dynamic array was implemented using a normalization factor (NF) that was calculated using the mRNA expression levels of three references genes (ACTB, HPRT1 and GAPDH), which were also measured in the same qPCR array. In particular, this NF was calculated using the GeNorm 3.3 software (Vandesompele, De Preter et al. 2002), which consider the expression of these housekeeping genes and provide a NF for each particular sample. Therefore, each sample was normalized using its NF in order to carry out a correctevaluation and interpretation of the data. Indeed, we have previously reported this method in other articles (Del Rio-Moreno, Alors-Perez et al. 2019, Vazquez-Borrego, L-Lopez et al. 2019). This information can be found in page 15 (lines 511-514).

Regarding the second part of the question, we have to underline that this seems to be, by itself, a valuable novel insight. Indeed, it is well known that PitNETs comprise a highly heterogenous pathology, inasmuch as each PitNET subtype arises from the expansion of a different hormonal cell type, and thereby, this results in that the behavior of each of PitNETs subtype, from the molecular, functional and clinical point of view, is highly different. This has been widely documented from many points of view (Asa and Ezzat 2009, Melmed 2011, Lim and Korbonits 2018). In this scenario, our present observation that the same splicing-machinery related transcripts display distinct expression levels when comparing the different PitNETs subtypes with normal pituitaries, used as a normal-control tissue for reference purposes, is not only not surprising, but provides a novel piece of evidence to support the highly and intrinsic heterogeneous nature of pituitary neuroendocrine tumors, and may therefore shed novel light on the molecular basis underlying this clinically-relevant heterogeneity.

Reviewer:The authors mention that some of the identified splicing factors are correlated with aberrant splicing variants, e.g. page 4, lines 162-164. However, the data are not shown.

Authors: We appreciate this very helpful comment by the Reviewer. Indeed, all the observed correlations between splicing factors and aberrant splicing variants found in this study have been specified throughout the manuscript (indicating the r- and p-values). Indeed, we describe that the splicing factor RAVER1 was significantly and positively correlated with the expression levels of the aberrant splicing variant In1-ghrelin, showing a Spearman’s Rho coefficient of r: 0.398 and p value of 0.024 in GHomas (pages 4-5; lines 193-196). In the same line, we found a positive correlation of the splicing factor MAGOH with the aberrant splicing variant SST5TMD4, showing a Spearman’s Rho coefficient of r: 0.491 and p value of 0.033 in ACTHomas (page 5; lines 220-222). Additionally, in the section 2.5 (“Similar dysregulation of specific splicing machinery components in all PitNET subtypes”), we described: 1) that SRSF1 mRNA levels were directly correlated with the aberrant splicing variant SST5TMD4 in NFPTs (r: 0.425; p= 0.049), GHomas (r: 0.532; p= 0.002) and ACTHomas (r: 0.509; p= 0.026), and; 2) that expression levels of SST5TMD4, but not SST5, were also directly associated with those of RNU11 (r: 0.392; p= 0.026), RNU4ATAC (r: 0.422; p= 0.016) and RNU6ATAC (r: 0.450; p= 0.010) in GHomas. In any case, we have revised the full manuscript to ensure that all the correlations have been appropriately indicated.

Reviewer: Figure 5: What is the difference between cell proliferation and cell viability assays? It seems to me that they are identical.

Authors: We thank the Reviewer for this important observation. In this regard, we would like to emphasize that cell proliferation and cell viability are not exactly the same, at least, as they are considered in this study. Indeed, cell proliferation is referred to an increase in cell number with time due to cell division, while cell viability refers to the number of living cells within a population. In our study, we strengthen this distinction since AtT-20 and GH3 pituitary cell lines are immortalized cells able to proliferate and measurably grow over time. In contrast, although human PitNETs are able to proliferate, in general, at a very low rate in a in vivosituation (in the patients), the primary cell cultures derived from PitNETs samples exhibit a very reduced (practically negligible) growth rate, probably due to lack of the complex environment existing in the in vivoconditions. For these reasons, as previously reported in different papers from our group (Vazquez-Borrego, Fuentes-Fayos et al. 2019), and in order to be more precise, we prefer to use the term cell viability for human primary PitNETs cell cultures and the term cell proliferation for PitNETs cell lines.

Reviewer:Figure 5: Please add a third replicate to Figure 5C to obtain a statistically relevant result for changes in GH secretion.

Authors: We thank the Reviewer for this pertinent comment. However, and as we mentioned above, the availability of tumor-derived samples with enough number of cells to implement this type of experiment is limited and that the time required for such experimentation would imply a long period of time (keeping in mind that the editorial regulations provided 10 days as the time limit to resubmit a revised version of the present manuscript). For these reasons, we would like to kindly request to the Reviewer that this would not be a mandatory requisite for acceptance of the manuscript.

Reviewer:Discussion, page 11 line 325: SRSF9 and SND1 are overexpressed in several tumors, but reduced in NFPTs. This contradictory result should be discussed.

Authors: We truly appreciate the comment of the Reviewer. First, we want to emphasize that in our study, SRSF9 and SND1 were significantly downregulated in NFPTs but overexpressed in GHomas, while they were not altered in the other PitNETs subtypes. Having said this, it is worth noting that the overexpression of these two splicing factors has been related with an increase of cell proliferation and poor prognosis in several tumor types such as breast cancer, bladder cancer or glioblastoma. Nevertheless, as we mentioned above, PitNETs, and specially NFPTs, are a very heterogeneous pathology that encompasses a great number of different behaviors. Indeed, NFPTs not only showed a downregulation of SRSF9 and SND1, but also exhibited a profound downregulation of most of the components of the splicing machinery analyzed in comparison with normal pituitary glands and other PitNETs subtypes. This is not surprising, since NFPTs are quite different to functioning PitNETs (GHomas, ACTHomas and PRLomas) and typically display a distinct behavior (often macroadenomas lacking hormone hypersecretion) and different expression pattern of relevant components involved in pituitary cell function, such as somatostatin receptors, in comparison with functioning PitNETs and normal tissue. For these reasons, in our opinion, the observed downregulation of these splicing factors is likely a reflection of the complexity, heterogeneity and limited functional deployment of these tumors. Nonetheless, future studies should be conducted to elucidate this notion and to better understand the role of these specific dysregulations in the functionality of this pathology. Accordingly, and based on the Reviewer’s comment, we have included this explanation in the Discussion section [page 12 (lines 353-361 and lines 373-376)], as follows: “One of the main findings of this study is the discovery that the spliceosome machinery is dysregulated in a tumor subtype-dependent manner, where NFPTs, GHomas, ACTHomas and PRLomas exhibit a differentially altered pattern of expression. Of particular interest are the results found in NFPTs, which displayed a profound downregulation of most of the components analyzed, in striking contrast with the alterations observed in functioning PitNETs (GHomas, ACTHomas and PRLomas), wherein many components are overexpressed. In line with this, previous results have demonstrated that NFPTs typically display a distinct behavior and different expression pattern of relevant components involved in pituitary cell function, such as somatostatin receptors, in comparison with functioning PitNETs and normal tissue [42-45].”; and “The fact that these SFs were downregulated in NFPTs, in contrast with the overexpression observed in other pathologies, may be likely reflect the complexity, heterogeneity and limited functional deployment of these tumors.”.

Reviewer:Figure S2B: Most of the tumors do not seem to express SST1 and SST5 mRNAs at all. What percentage of tumor samples expressed these two mRNAs? Is there still a significant difference if only the samples expressing the mRNAs are considered? Why are these mRNA levels not normalized by the “normalization factor”, but by ACTB mRNA levels?

Authors: We understand the concern of the Reviewer about SSTs expression levels. In this sense, we first want to clarify that the mRNA expression levels of these two SSTs is much lower in NFPTs compared to the expression of SST3and SST2(Taboada, Luque et al. 2007, Ibanez-Costa, Rivero-Cortes et al. 2016). Having said this, all NFPTs analyzed expressed SST1and only the 11.4% of tumors did not express SST5. The rest of the tumors analyzed expressed very low or moderate expression levels of these receptors, and therefore the results are practically the same if only the samples expressing these receptors are considered. Indeed, the analysis of the mRNA levels of SST5taking in account only the tumors expressing this receptor still showed a significant difference (p=0.0038) between NFPTs P1 and P2. Finally, we have to clarify that in the case of the splicing machinery, all the transcripts were normalized by the normalization factor (NF) calculated with three reference genes (ACTB, HPRT1 and GAPDH), since the configuration of the array allowed us to include those reference genes (using a very low amount of samples). However, due to the limited amount of sample available, we were only able to use one reference gene to analyze the results from the conventional qPCR. In this sense, we evaluated the stability of the expression of three reference genes ACTB, HPRT1 and GAPDH in all samples using RefFinder (Xie, Xiao et al. 2012), a comprehensive tool that integrates the currently available major computational programmes, and found ACTB to be the most stable.

Reviewer:Figure S6B: A significant reduction of RNU6ATAC and SRSF1 is observed only in NFPTs, but not in the other three subclasses. This can hardly be referred to as a "common fingerprint". Why, for example, was SRRM4 induction not considered? It reaches statistical significance in three out of four subclasses?

Authors: Regarding this question, it is true that RNU6ATAC and SRSF1 reached statistically significance only in NFPTs. However, we considered that these results are worth to mention since, despite not reaching statistical significance, the dysregulation that these two elements appeared to show in the rest of PitNETs subtypes was also visually noticeable. Indeed, it is important to take into account that the number of samples included in the analysis is much higher in NFPTs (n=88) compared to functioning PitNETs (48 GHomas, 22 ACTHomas and 7 PRLomas). On the other hand, we agree with the Reviewer in that the overexpression of SRRM4 in three out of four PitNETs subtypes analyzed is very relevant and we appreciate this suggestion. Therefore, we have modified the figure S7A-B to indicate also the dysregulation of SRRM4 as a common alteration between the different PitNETs subtypes, which might be patho-physiologically relevant for the better understanding of this pathology. Thus, we have included this information in pages 6 (line 243), 14 (line 438) and 17 (line 605), and in supplemental figure S7A-B.

REFERENCES

Asa, S. L. and S. Ezzat (2009). "The pathogenesis of pituitary tumors." Annu Rev Pathol4: 97-126.

Del Rio-Moreno, M., E. Alors-Perez, S. Gonzalez-Rubio, G. Ferrin, O. Reyes, M. Rodriguez-Peralvarez, M. E. Sanchez-Frias, R. Sanchez-Sanchez, S. Ventura, J. Lopez-Miranda, R. D. Kineman, M. de la Mata, J. P. Castano, M. D. Gahete and R. M. Luque (2019). "Dysregulation of the Splicing Machinery Is Associated to the Development of Nonalcoholic Fatty Liver Disease." J Clin Endocrinol Metab104(8): 3389-3402.

Ibanez-Costa, A., E. Rivero-Cortes, M. C. Vazquez-Borrego, M. D. Gahete, L. Jimenez-Reina, E. Venegas-Moreno, A. de la Riva, M. A. Arraez, I. Gonzalez-Molero, H. A. Schmid, S. Maraver-Selfa, I. Gavilan-Villarejo, J. A. Garcia-Arnes, M. A. Japon, A. Soto-Moreno, M. A. Galvez, R. M. Luque and J. P. Castano (2016). "Octreotide and pasireotide (dis)similarly inhibit pituitary tumor cells in vitro." J Endocrinol231(2): 135-145.

Lim, C. T. and M. Korbonits (2018). "Update on the Clinicopathology of Pituitary Adenomas." Endocr Pract24(5): 473-488.

Melmed, S. (2011). "Pathogenesis of pituitary tumors." Nat Rev Endocrinol7(5): 257-266.

Taboada, G. F., R. M. Luque, W. Bastos, R. F. Guimaraes, J. B. Marcondes, L. M. Chimelli, R. Fontes, P. J. Mata, P. N. Filho, D. P. Carvalho, R. D. Kineman and M. R. Gadelha (2007). "Quantitative analysis of somatostatin receptor subtype (SSTR1-5) gene expression levels in somatotropinomas and non-functioning pituitary adenomas." Eur J Endocrinol156(1): 65-74.

Vandesompele, J., K. De Preter, F. Pattyn, B. Poppe, N. Van Roy, A. De Paepe and F. Speleman (2002). "Accurate normalization of real-time quantitative RT-PCR data by geometric averaging of multiple internal control genes." Genome Biol3(7): RESEARCH0034.

Vazquez-Borrego, M. C., A. C. Fuentes-Fayos, A. D. Herrera-Martinez, F. L-Lopez, A. Ibanez-Costa, P. Moreno-Moreno, M. R. Alhambra-Exposito, A. Barrera-Martin, C. Blanco-Acevedo, E. Dios, E. Venegas-Moreno, J. Solivera, M. D. Gahete, A. Soto-Moreno, M. A. Galvez-Moreno, J. P. Castano and R. M. Luque (2019). "Biguanides Exert Antitumoral Actions in Pituitary Tumor Cells Through AMPK-Dependent and -Independent Mechanisms." The Journal of clinical endocrinology and metabolism104(8): 3501-3513.

Vazquez-Borrego, M. C., F. L-Lopez, M. A. Galvez-Moreno, A. C. Fuentes-Fayos, E. Venegas-Moreno, A. D. Herrera-Martinez, C. Blanco-Acevedo, J. Solivera, T. Landsman, M. D. Gahete, A. Soto-Moreno, M. D. Culler, J. P. Castano and R. M. Luque (2019). "A new generation somatostatin-dopamine analogue exerts potent antitumoral actions on pituitary neuroendocrine tumor cells." Neuroendocrinology. [Epub ahead of print].

Xie, F., P. Xiao, D. Chen, L. Xu and B. Zhang (2012). "miRDeepFinder: a miRNA analysis tool for deep sequencing of plant small RNAs." Plant Mol Biol.

Reviewer 3 Report

This paper by Vázquez-Borrego and collaborators reports results on the expression dysregulation of spliceosomal components in pituitary neuroendocrine tumors. The authors showed the pattern of expression of the splicing machinery in the main types of pituitary tumors by a custom qPCR dynamic array. They provided evidence that spliceosome machinery is dysregulated in a tumor subtype-dependent manner, while a common pattern of downregulation of two minor spliceosome components is present. Finally, although preliminarily, they clearly showed the potential beneficial effect of pladienolide-B in the treatment of pituitary neuroendocrine tumors.  
Overall, the manuscript is well written, and the experimental design is convincing and well reported.

Minor point

The authors should better document the rationale behind the used of pladienolide-B in this study, considering that SF3B1 did not result dysregulated in their qPCR dynamic array

The paper will benefit from wound healing analysis on AtT-20 corticotrophs and GH3 somatotrophs cell lines after pladienolide-B.

Author Response

RESPONSES TO REVIEWER #3:

Reviewer: The authors should better document the rationale behind the used of pladienolide-B in this study, considering that SF3B1 did not result dysregulated in their qPCR dynamic array.

Authors:We thank again the Reviewer for this pertinent comment. In this regard, the main reason by which we used this compound, even if SF3B1 is not dysregulated in our samples, is based on that the drugs available to target the splicing machinery are very limited and the majority of them are targeted to SF3B1. SF3B1 is a principal player in the spliceosome and the binding of pladienolide-B to SF3B1 produces the destabilization of the recruitment of snRNP U2 and the inhibition of the splicing process. In fact, this compound has been reported to inhibit the splicing process in several tumor cells (Sato, Muguruma et al. 2014, Wu, Fan et al. 2018, Zhang, Di et al. 2019). For these reasons, we used this compound due to its capacity to naturally disrupt the splicing process targeting SF3B1 regardless its target was not dysregulated in our samples of PitNETs. This information can be found in page 14 (lines 455-460).

Reviewer: The paper will benefit from wound healing analysis on AtT-20 corticotrophs and GH3 somatotrophs cell lines after pladienolide-B.

Authors: We truly appreciate this comment. However, in our extended experience, performing reliable and reproducible wound healing assays is not feasible in AtT-20 and GH3 cell lines, due to the tendency of these cells to grow in aggregates rather than in a monolayer, which is essential to perform a correct wound healing analysis. In fact, we have tried to perform this type of experiment several times for previous articles in which we used these cell lines but we were unable to obtain reliable results (Vazquez-Borrego, Fuentes-Fayos et al. 2019, Vazquez-Borrego, L-Lopez et al. 2019). For these reasons, we did not include this type of experiment in the present study.

REFERENCES

Sato, M., N. Muguruma, T. Nakagawa, K. Okamoto, T. Kimura, S. Kitamura, H. Yano, K. Sannomiya, T. Goji, H. Miyamoto, T. Okahisa, H. Mikasa, S. Wada, M. Iwata and T. Takayama (2014). "High antitumor activity of pladienolide B and its derivative in gastric cancer." Cancer science105(1): 110-116.

Vazquez-Borrego, M. C., A. C. Fuentes-Fayos, A. D. Herrera-Martinez, F. L-Lopez, A. Ibanez-Costa, P. Moreno-Moreno, M. R. Alhambra-Exposito, A. Barrera-Martin, C. Blanco-Acevedo, E. Dios, E. Venegas-Moreno, J. Solivera, M. D. Gahete, A. Soto-Moreno, M. A. Galvez-Moreno, J. P. Castano and R. M. Luque (2019). "Biguanides Exert Antitumoral Actions in Pituitary Tumor Cells Through AMPK-Dependent and -Independent Mechanisms." The Journal of clinical endocrinology and metabolism104(8): 3501-3513.

Vazquez-Borrego, M. C., F. L-Lopez, M. A. Galvez-Moreno, A. C. Fuentes-Fayos, E. Venegas-Moreno, A. D. Herrera-Martinez, C. Blanco-Acevedo, J. Solivera, T. Landsman, M. D. Gahete, A. Soto-Moreno, M. D. Culler, J. P. Castano and R. M. Luque (2019). "A new generation somatostatin-dopamine analogue exerts potent antitumoral actions on pituitary neuroendocrine tumor cells." Neuroendocrinology. [Epub ahead of print].

Wu, G., L. Fan, M. N. Edmonson, T. Shaw, K. Boggs, J. Easton, M. C. Rusch, T. R. Webb, J. Zhang and P. M. Potter (2018). "Corrigendum: Inhibition of SF3B1 by molecules targeting the spliceosome results in massive aberrant exon skipping." RNA (New York, N Y )24(12): 1886.

Zhang, Q., C. Di, J. Yan, F. Wang, T. Qu, Y. Wang, Y. Chen, X. Zhang, Y. Liu, H. Yang and H. Zhang (2019). "Inhibition of SF3b1 by pladienolide B evokes cycle arrest, apoptosis induction and p73 splicing in human cervical carcinoma cells." Artificial cells, nanomedicine, and biotechnology47(1): 1273-1280.

Round 2

Reviewer 1 Report

The authors extensively addressed all my concerns and provided an improved version of their work. 

Reviewer 2 Report

Author's Notes

RESPONSES TO REVIEWER #2:

Reviewer:The widespread dysregulation of the splicing machinery has already been extensively documented in a large number of cancers. Thus, while findings on the dysregulation of the splicing machinery in cancer may translate into novel therapeutic approaches and as such would be of interest to a broad readership, this manuscript is not sufficient to provide significant new insights.Widespread dysregulation of the splicing machinery has been amply documented in a plethora of cancer types.

Authors: We agree with the Reviewer in that different studies have demonstrated that certain splicing machinery components are dysregulated in several cancer types, which may help to anticipate that the general splicing machinery could be altered in a large number of cancers. However, to the best of our knowledge, the present approach, focused on the analysis of the expression of a relevant number of representative splicing machinery, and performed in parallel in a representative set of tumor samples, has not been previously reported. Moreover, as the Reviewer surely knows, tumor pathologies and cancers are very heterogeneous entities that have to be analyzed independently in order to unveil their particular features. Thus, in most studies investigating splicing in different cancer types, an analysis directed to assess the expression of the main components of the splicing machinery has not been the main focus, and thus, earlier approaches did not explore in detail the dysregulation of the splicing machinery in most cancer types. In this sense, we have implemented herein, for the first time and by the use of a pioneer, splicing machinery-focused and, custom-designed and highly sensitive approach, a comprehensive and comparative analysis of a relevant number of representative splicing machinery elements in PitNETs, demonstrating that a high proportion of these elements (including spliceosome components and splicing factors) are differentially dysregulated in different PitNETs subtypes. Therefore, we humbly consider that this study provides significant novel insights to the current state of the art and opens new conceptual pathways and methodological approaches for the study of these and other tumor pathologies.

Reviewer:The authors show quantification of mRNA level changes for 42 spliceosome components. Using different clustering algorithms, they define the most significant changes, which allow separation of tumor cells from normal tissue. Further, they show correlation for some of these factors with clinical parameters. However, the authors fail to show that the identified expression changes impact on cancer cell survival. In addition, they provide no mechanistic insights, such as downstream effects on gene expression caused directly by the reduction or induction of the identified splicing factors.

Authors: We appreciate this constructive criticism of the Reviewer. We can only concur that the impact of the splicing machinery dysregulation on cancer cell survival as well as the mechanistic insights behind this dysregulation comprise highly interesting subjects, which certainly deserve detailed investigation. Notwithstanding this, we have to point out that the main objective of our study was to implement, for the first time, a comprehensive and comparative characterization of the pattern of expression the splicing machinery components in different PitNETs subtypes compared to normal pituitaries, which unveiled a profound dysregulation of this pivotal machinery, identified novel altered targets and opened new avenues of research. In this sense, we should like to underscore that the obtaining of the high number of tumor samples from all PitNETs subtypes (being some of them very rare) analyzed in our study has been a very challenging endeavor, which required a long period of time (several years) and a tight, highly coordinated collaboration among different research centers. In addition, it should also be noted that the number of cells obtained for primary cultures after tumor dispersion is very limited, since the tumor size is usually small and the piece of tissue available for experimental studies is necessarily even smaller. For this reason, the (needless is to say) exciting and interesting study of the impact of the splicing machinery dysregulation on pituitary tumor cell survival as well as the mechanistic insights behind this dysregulation will be very challenging. Therefore, herein, as a general proof-of-concept, we had to decide to solely explore the consequences of the global inhibition of the splicing machinery by the use of pladienolide-B, which reduces SF3B1 activity and hampers the normal function of the spliceosome. Nevertheless, and in agreement with the comment by the Reviewer, we have planned to explore in the coming future the effect of the dysregulation of selected splicing machinery components on specific functional endpoints, as well as their underlying mechanistic insights. However, these planned studies will necessarily require several months and, likely, years of additional experimentation (and associated extra funding) and are meant to be the subject of independent manuscripts in the future. Accordingly, we would respectfully request to the Reviewer and Editor this additional novel and challenging data not to be considered as a requisite for the potential acceptance of our present revised manuscript.

I really appreciate that the collection and analysis of such a large number of tumor samples is very labor-intensive and time-consuming. I also appreciate that the acquisition of primary cultures is challenging. However, the authors themselves use cell lines to illustrate the effect of Pladienolide-B on pituitary tumors. As the authors show, these cell lines are very sensitive to the interference with the splicing machinery and could therefore be used as an accessible model system to investigate the effects of changes in certain splicing factors. I do not see how, for example, an siRNA-mediated knockdown of a splicing factor and a subsequent cell proliferation assay will take years. However, it will substantially strengthen the results of the study. For such an experiment, I would strongly suggest giving enough time for a second round of revisions.

Reviewer: The results section should start with a more comprehensive description of the experimental setup. For example, which and how many tumor samples were compared with which and how many control samples?

Authors: We thank again the Reviewer for this comment. In this regard, we have included a paragraph at the beginning of the result section explaining in detail all the samples and the splicing machinery components analyzed in our study. This information can be found in page 3 (lines 102-109). “In the present study, we analyzed simultaneously the expression levels of 42 components of the splicing machinery (12 components of major spliceosome, 4 components of the minor spliceosome and 26 SFs) in different PitNETs subtypes using a dynamic quantitative real-time PCR (qPCR) microfluidic array. Specifically, we evaluated the dysregulations of these spliceosome components and SFs in an ample range of human PitNETs samples in comparison with NP-glands. Thus, we analyze 88 NFPTs, 48 GHomas, 22 ACTHomas, 7 PRLomas and 11 NPs (Cohort from Spain). Additionally, we had the opportunity to evaluate the dysregulation of the splicing machinery in a second cohort of 83 GHomas from Brazil.”.

Reviewer:Of particular interest is the normalization of qPCR data. In the Material and Methods section, the authors state that three housekeeping genes were measured. Figure legends state that the data are "adjusted by a normalization factor". However, what this factor is and how normalization was performed is not described. However, this information is crucial for the evaluation and interpretation of the validity of the data. On this note, how is it possible that the same transcripts produce different results in the control compared to different PitNET subclasses (e.g. Figure S6)?

Authors: We sincerely appreciate this pertinent observation and apologize for not being clearer in this aspect. Specifically,the normalization of the expression of all the transcripts determined in the qPCR dynamic array was implemented using a normalization factor (NF) that was calculated using the mRNA expression levels of three references genes (ACTB, HPRT1 and GAPDH), which were also measured in the same qPCR array. In particular, this NF was calculated using the GeNorm 3.3 software (Vandesompele, De Preter et al. 2002), which consider the expression of these housekeeping genes and provide a NF for each particular sample. Therefore, each sample was normalized using its NF in order to carry out a correctevaluation and interpretation of the data. Indeed, we have previously reported this method in other articles (Del Rio-Moreno, Alors-Perez et al. 2019, Vazquez-Borrego, L-Lopez et al. 2019). This information can be found in page 15 (lines 511-514).

Regarding the second part of the question, we have to underline that this seems to be, by itself, a valuable novel insight. Indeed, it is well known that PitNETs comprise a highly heterogenous pathology, inasmuch as each PitNET subtype arises from the expansion of a different hormonal cell type, and thereby, this results in that the behavior of each of PitNETs subtype, from the molecular, functional and clinical point of view, is highly different. This has been widely documented from many points of view (Asa and Ezzat 2009, Melmed 2011, Lim and Korbonits 2018). In this scenario, our present observation that the same splicing-machinery related transcripts display distinct expression levels when comparing the different PitNETs subtypes with normal pituitaries, used as a normal-control tissue for reference purposes, is not only not surprising, but provides a novel piece of evidence to support the highly and intrinsic heterogeneous nature of pituitary neuroendocrine tumors, and may therefore shed novel light on the molecular basis underlying this clinically-relevant heterogeneity.

I think there has been a misunderstanding. I asked why the control shows different levels for the same transcript. I assumed that the same 11 NP control samples were compared with the different PitNET subtypes. Then how is it possible that e.g. the level for RNU6ATAC is 1,000,000 for the control when compared to ACTHomas and 3,000,000 when compared to PRLomas (now Figure S7)?

Reviewer:The authors mention that some of the identified splicing factors are correlated with aberrant splicing variants, e.g. page 4, lines 162-164. However, the data are not shown.

Authors: We appreciate this very helpful comment by the Reviewer. Indeed, all the observed correlations between splicing factors and aberrant splicing variants found in this study have been specified throughout the manuscript (indicating the r- and p-values). Indeed, we describe that the splicing factor RAVER1 was significantly and positively correlated with the expression levels of the aberrant splicing variant In1-ghrelin, showing a Spearman’s Rho coefficient of r: 0.398 and p value of 0.024 in GHomas (pages 4-5; lines 193-196). In the same line, we found a positive correlation of the splicing factor MAGOH with the aberrant splicing variant SST5TMD4, showing a Spearman’s Rho coefficient of r: 0.491 and p value of 0.033 in ACTHomas (page 5; lines 220-222). Additionally, in the section 2.5 (“Similar dysregulation of specific splicing machinery components in all PitNET subtypes”), we described: 1) that SRSF1 mRNA levels were directly correlated with the aberrant splicing variant SST5TMD4 in NFPTs (r: 0.425; p= 0.049), GHomas (r: 0.532; p= 0.002) and ACTHomas (r: 0.509; p= 0.026), and; 2) that expression levels of SST5TMD4, but not SST5, were also directly associated with those of RNU11 (r: 0.392; p= 0.026), RNU4ATAC (r: 0.422; p= 0.016) and RNU6ATAC (r: 0.450; p= 0.010) in GHomas. In any case, we have revised the full manuscript to ensure that all the correlations have been appropriately indicated.

I would like to know how these alternative splicing events have been quantified. Could you visualize these correlations in a graph? A short description would also be helpful. For example, what is "In1-ghrelin"?

Reviewer: Figure 5: What is the difference between cell proliferation and cell viability assays? It seems to me that they are identical.

Authors: We thank the Reviewer for this important observation. In this regard, we would like to emphasize that cell proliferation and cell viability are not exactly the same, at least, as they are considered in this study. Indeed, cell proliferation is referred to an increase in cell number with time due to cell division, while cell viability refers to the number of living cells within a population. In our study, we strengthen this distinction since AtT-20 and GH3 pituitary cell lines are immortalized cells able to proliferate and measurably grow over time. In contrast, although human PitNETs are able to proliferate, in general, at a very low rate in a in vivosituation (in the patients), the primary cell cultures derived from PitNETs samples exhibit a very reduced (practically negligible) growth rate, probably due to lack of the complex environment existing in the in vivoconditions. For these reasons, as previously reported in different papers from our group (Vazquez-Borrego, Fuentes-Fayos et al. 2019), and in order to be more precise, we prefer to use the term cell viability for human primary PitNETs cell cultures and the term cell proliferation for PitNETs cell lines.

Reviewer:Figure 5: Please add a third replicate to Figure 5C to obtain a statistically relevant result for changes in GH secretion.

Authors: We thank the Reviewer for this pertinent comment. However, and as we mentioned above, the availability of tumor-derived samples with enough number of cells to implement this type of experiment is limited and that the time required for such experimentation would imply a long period of time (keeping in mind that the editorial regulations provided 10 days as the time limit to resubmit a revised version of the present manuscript). For these reasons, we would like to kindly request to the Reviewer that this would not be a mandatory requisite for acceptance of the manuscript.

Reviewer:Discussion, page 11 line 325: SRSF9 and SND1 are overexpressed in several tumors, but reduced in NFPTs. This contradictory result should be discussed.

Authors: We truly appreciate the comment of the Reviewer. First, we want to emphasize that in our study, SRSF9 and SND1 were significantly downregulated in NFPTs but overexpressed in GHomas, while they were not altered in the other PitNETs subtypes. Having said this, it is worth noting that the overexpression of these two splicing factors has been related with an increase of cell proliferation and poor prognosis in several tumor types such as breast cancer, bladder cancer or glioblastoma. Nevertheless, as we mentioned above, PitNETs, and specially NFPTs, are a very heterogeneous pathology that encompasses a great number of different behaviors. Indeed, NFPTs not only showed a downregulation of SRSF9 and SND1, but also exhibited a profound downregulation of most of the components of the splicing machinery analyzed in comparison with normal pituitary glands and other PitNETs subtypes. This is not surprising, since NFPTs are quite different to functioning PitNETs (GHomas, ACTHomas and PRLomas) and typically display a distinct behavior (often macroadenomas lacking hormone hypersecretion) and different expression pattern of relevant components involved in pituitary cell function, such as somatostatin receptors, in comparison with functioning PitNETs and normal tissue. For these reasons, in our opinion, the observed downregulation of these splicing factors is likely a reflection of the complexity, heterogeneity and limited functional deployment of these tumors. Nonetheless, future studies should be conducted to elucidate this notion and to better understand the role of these specific dysregulations in the functionality of this pathology. Accordingly, and based on the Reviewer’s comment, we have included this explanation in the Discussion section [page 12 (lines 353-361 and lines 373-376)], as follows: “One of the main findings of this study is the discovery that the spliceosome machinery is dysregulated in a tumor subtype-dependent manner, where NFPTs, GHomas, ACTHomas and PRLomas exhibit a differentially altered pattern of expression. Of particular interest are the results found in NFPTs, which displayed a profound downregulation of most of the components analyzed, in striking contrast with the alterations observed in functioning PitNETs (GHomas, ACTHomas and PRLomas), wherein many components are overexpressed. In line with this, previous results have demonstrated that NFPTs typically display a distinct behavior and different expression pattern of relevant components involved in pituitary cell function, such as somatostatin receptors, in comparison with functioning PitNETs and normal tissue [42-45].”; and “The fact that these SFs were downregulated in NFPTs, in contrast with the overexpression observed in other pathologies, may be likely reflect the complexity, heterogeneity and limited functional deployment of these tumors.”.

Reviewer:Figure S2B: Most of the tumors do not seem to express SST1 and SST5 mRNAs at all. What percentage of tumor samples expressed these two mRNAs? Is there still a significant difference if only the samples expressing the mRNAs are considered? Why are these mRNA levels not normalized by the “normalization factor”, but by ACTB mRNA levels?

Authors: We understand the concern of the Reviewer about SSTs expression levels. In this sense, we first want to clarify that the mRNA expression levels of these two SSTs is much lower in NFPTs compared to the expression of SST3and SST2(Taboada, Luque et al. 2007, Ibanez-Costa, Rivero-Cortes et al. 2016). Having said this, all NFPTs analyzed expressed SST1and only the 11.4% of tumors did not express SST5. The rest of the tumors analyzed expressed very low or moderate expression levels of these receptors, and therefore the results are practically the same if only the samples expressing these receptors are considered. Indeed, the analysis of the mRNA levels of SST5taking in account only the tumors expressing this receptor still showed a significant difference (p=0.0038) between NFPTs P1 and P2. Finally, we have to clarify that in the case of the splicing machinery, all the transcripts were normalized by the normalization factor (NF) calculated with three reference genes (ACTB, HPRT1 and GAPDH), since the configuration of the array allowed us to include those reference genes (using a very low amount of samples). However, due to the limited amount of sample available, we were only able to use one reference gene to analyze the results from the conventional qPCR. In this sense, we evaluated the stability of the expression of three reference genes ACTB, HPRT1 and GAPDH in all samples using RefFinder (Xie, Xiao et al. 2012), a comprehensive tool that integrates the currently available major computational programmes, and found ACTB to be the most stable.

Reviewer:Figure S6B: A significant reduction of RNU6ATAC and SRSF1 is observed only in NFPTs, but not in the other three subclasses. This can hardly be referred to as a "common fingerprint". Why, for example, was SRRM4 induction not considered? It reaches statistical significance in three out of four subclasses?

Authors: Regarding this question, it is true that RNU6ATAC and SRSF1 reached statistically significance only in NFPTs. However, we considered that these results are worth to mention since, despite not reaching statistical significance, the dysregulation that these two elements appeared to show in the rest of PitNETs subtypes was also visually noticeable. Indeed, it is important to take into account that the number of samples included in the analysis is much higher in NFPTs (n=88) compared to functioning PitNETs (48 GHomas, 22 ACTHomas and 7 PRLomas). On the other hand, we agree with the Reviewer in that the overexpression of SRRM4 in three out of four PitNETs subtypes analyzed is very relevant and we appreciate this suggestion. Therefore, we have modified the figure S7A-B to indicate also the dysregulation of SRRM4 as a common alteration between the different PitNETs subtypes, which might be patho-physiologically relevant for the better understanding of this pathology. Thus, we have included this information in pages 6 (line 243), 14 (line 438) and 17 (line 605), and in supplemental figure S7A-B.

REFERENCES

Asa, S. L. and S. Ezzat (2009). "The pathogenesis of pituitary tumors." Annu Rev Pathol4: 97-126.

Del Rio-Moreno, M., E. Alors-Perez, S. Gonzalez-Rubio, G. Ferrin, O. Reyes, M. Rodriguez-Peralvarez, M. E. Sanchez-Frias, R. Sanchez-Sanchez, S. Ventura, J. Lopez-Miranda, R. D. Kineman, M. de la Mata, J. P. Castano, M. D. Gahete and R. M. Luque (2019). "Dysregulation of the Splicing Machinery Is Associated to the Development of Nonalcoholic Fatty Liver Disease." J Clin Endocrinol Metab104(8): 3389-3402.

Ibanez-Costa, A., E. Rivero-Cortes, M. C. Vazquez-Borrego, M. D. Gahete, L. Jimenez-Reina, E. Venegas-Moreno, A. de la Riva, M. A. Arraez, I. Gonzalez-Molero, H. A. Schmid, S. Maraver-Selfa, I. Gavilan-Villarejo, J. A. Garcia-Arnes, M. A. Japon, A. Soto-Moreno, M. A. Galvez, R. M. Luque and J. P. Castano (2016). "Octreotide and pasireotide (dis)similarly inhibit pituitary tumor cells in vitro." J Endocrinol231(2): 135-145.

Lim, C. T. and M. Korbonits (2018). "Update on the Clinicopathology of Pituitary Adenomas." Endocr Pract24(5): 473-488.

Melmed, S. (2011). "Pathogenesis of pituitary tumors." Nat Rev Endocrinol7(5): 257-266.

Taboada, G. F., R. M. Luque, W. Bastos, R. F. Guimaraes, J. B. Marcondes, L. M. Chimelli, R. Fontes, P. J. Mata, P. N. Filho, D. P. Carvalho, R. D. Kineman and M. R. Gadelha (2007). "Quantitative analysis of somatostatin receptor subtype (SSTR1-5) gene expression levels in somatotropinomas and non-functioning pituitary adenomas." Eur J Endocrinol156(1): 65-74.

Vandesompele, J., K. De Preter, F. Pattyn, B. Poppe, N. Van Roy, A. De Paepe and F. Speleman (2002). "Accurate normalization of real-time quantitative RT-PCR data by geometric averaging of multiple internal control genes." Genome Biol3(7): RESEARCH0034.

Vazquez-Borrego, M. C., A. C. Fuentes-Fayos, A. D. Herrera-Martinez, F. L-Lopez, A. Ibanez-Costa, P. Moreno-Moreno, M. R. Alhambra-Exposito, A. Barrera-Martin, C. Blanco-Acevedo, E. Dios, E. Venegas-Moreno, J. Solivera, M. D. Gahete, A. Soto-Moreno, M. A. Galvez-Moreno, J. P. Castano and R. M. Luque (2019). "Biguanides Exert Antitumoral Actions in Pituitary Tumor Cells Through AMPK-Dependent and -Independent Mechanisms." The Journal of clinical endocrinology and metabolism104(8): 3501-3513.

Vazquez-Borrego, M. C., F. L-Lopez, M. A. Galvez-Moreno, A. C. Fuentes-Fayos, E. Venegas-Moreno, A. D. Herrera-Martinez, C. Blanco-Acevedo, J. Solivera, T. Landsman, M. D. Gahete, A. Soto-Moreno, M. D. Culler, J. P. Castano and R. M. Luque (2019). "A new generation somatostatin-dopamine analogue exerts potent antitumoral actions on pituitary neuroendocrine tumor cells." Neuroendocrinology. [Epub ahead of print].

Xie, F., P. Xiao, D. Chen, L. Xu and B. Zhang (2012). "miRDeepFinder: a miRNA analysis tool for deep sequencing of plant small RNAs." Plant Mol Biol.